# Nucleo-cytoplasmic shuttling of splicing factor SRSF1 is required for development and cilia function

Fiona Haward[1†‡], Magdalena M Maslon[1†], Patricia L Yeyati[1†], Nicolas Bellora[2], Jan N Hansen[3], Stuart Aitken[1], Jennifer Lawson[1], Alex von Kriegsheim[4], Dagmar Wachten[3], Pleasantine Mill[1*], Ian R Adams[1*], Javier F Caceres[1*]

[1]MRC Human Genetics Unit, Institute of Genetics and Cancer, University of Edinburgh, Edinburgh, United Kingdom; [2]Institute of Nuclear Technologies for Health (Intecnus), National Scientific and Technical Research Council (CONICET), Bariloche, Argentina; [3]Institute of Innate Immunity, Biophysical Imaging, Medical Faculty, University of Bonn, Bonn, Germany; [4]Edinburgh Cancer Research United Kingdom Centre, Institute of Genetics and Cancer, University of Edinburgh, Edinburgh, United Kingdom

**\*For correspondence:**
Plesantine.Mill@ed.ac.uk (PM);
Ian.Adams@ed.ac.uk (IRA);
Javier.Caceres@ed.ac.uk (JFC)

[†]These authors contributed equally to this work

**Present address:** [‡] Centre for Gene Regulation and Expression, School of Life, University of Dundee, Dundee, United kingdom

**Competing interests:** The authors declare that no competing interests exist.

**Abstract** Shuttling RNA-binding proteins coordinate nuclear and cytoplasmic steps of gene expression. The SR family proteins regulate RNA splicing in the nucleus and a subset of them, including SRSF1, shuttles between the nucleus and cytoplasm affecting post-splicing processes. However, the physiological significance of this remains unclear. Here, we used genome editing to knock-in a nuclear retention signal (NRS) in *Srsf1* to create a mouse model harboring an SRSF1 protein that is retained exclusively in the nucleus. *Srsf1*[NRS/NRS] mutants displayed small body size, hydrocephalus, and immotile sperm, all traits associated with ciliary defects. We observed reduced translation of a subset of mRNAs and decreased abundance of proteins involved in multiciliogenesis, with disruption of ciliary ultrastructure and motility in cells and tissues derived from this mouse model. These results demonstrate that SRSF1 shuttling is used to reprogram gene expression networks in the context of high cellular demands, as observed here, during motile ciliogenesis.

## Introduction

Alternative splicing (AS) is an essential step in the gene expression cascade that generates a vast number of mRNA isoforms to shape the proteomes of multicellular eukaryotes (*Baralle and Giudice, 2017*; *Nilsen and Graveley, 2010*; *Ule and Blencowe, 2019*). It is largely controlled by the binding of RNA-binding proteins (RBPs) in a manner dependent on the cellular context (*Fu and Ares, 2014*). Additional layers of regulation include alterations in chromatin state and the co-transcriptional nature of the splicing process, including the rate of RNA Polymerase II (RNAPII) elongation (*Maslon et al., 2019*; *Naftelberg et al., 2015*; *Saldi et al., 2016*). Splicing alterations are found in human disease and are particularly common in cancer due to mutations in cis-acting splicing-regulatory elements or in components of the splicing machinery (*Anczuków and Krainer, 2016*; *Bonnal et al., 2020*; *Zhang and Manley, 2013*).

The serine/arginine-rich (SR) family proteins are among the most extensively characterized regulators of pre-mRNA splicing (*Wegener and Müller-McNicoll, 2019*; *Zhou et al., 2013*). Their modular domain structure comprises one or two RNA recognition motifs (RRMs) at their N-termini that contribute to sequence-specific RNA-binding and a C-terminal RS domain (arginine and serine repeats) that promotes protein-protein interactions (*Howard and Sanford, 2015*). The RS domain also acts

as a nuclear localization signal by promoting the interaction with the SR protein nuclear import receptor, transportin-SR (*Cáceres et al., 1997*; *Kataoka et al., 1999*; *Lai et al., 2000*). A combinatorial control mediated by both SR proteins and additional factors such as heterogeneous nuclear ribonucleoproteins (hnRNPs) influences AS patterns that generate specific cell lineages and tissues during development (*Busch and Hertel, 2012*; *Hanamura et al., 1998*; *Zhu et al., 2001*).

Besides clearly defined nuclear roles of SR family proteins, a subset of these, of which SRSF1 is the prototype, shuttle continuously from the nucleus to the cytoplasm and are involved in post-splicing activities (*Caceres et al., 1998*; *Cowper et al., 2001*; *Sapra et al., 2009*). These include mRNA export as well as cytoplasmic roles such as mRNA translation, nonsense-mediated decay (NMD), and regulation of RNA stability (reviewed by *Long and Caceres, 2009*; *Twyffels et al., 2011*; *Wagner and Frye, 2021*). Several protein kinases phosphorylate the RS domain of SR proteins, including the SRPK family (*Gui et al., 1994*) and the Clk/Sty family dual-specificity kinases (*Prasad et al., 1999*). Whereas phosphorylated SR proteins are required for spliceosome assembly, subsequent dephosphorylation is required for splicing catalysis and for sorting shuttling and non-shuttling SR proteins in the nucleus (*Lin et al., 2005*). Our previous work demonstrated that hypo-phosphorylated SRSF1 is associated with polyribosomes and promotes translation of mRNAs in an mTOR-dependent manner (*Michlewski et al., 2008*; *Sanford et al., 2005*; *Sanford et al., 2004*). We subsequently showed that translational targets of SRSF1 predominantly encode proteins that localize to centrosomes and are required for cell cycle regulation and chromosome segregation (*Maslon et al., 2014*). The shuttling ability of individual SR proteins has also been usurped by viruses, as seen with the role of individual SR proteins in promoting translation of viral transcripts, including poliovirus, MMPV, and HIV (*Bedard et al., 2007*; *Swartz et al., 2007*).

SRSF1 is essential for cellular viability (*Lin et al., 2005*; *Wang et al., 1996*) and has been shown to act as an oncogene and to promote mammary epithelial cell transformation (*Anczuków et al., 2012*; *Karni et al., 2007*). Targeted disruption of *Srsf1* in the mouse results in early embryonic lethality that cannot be rescued by other SR proteins (*Moroy and Heyd, 2007*; *Xu et al., 2005*). Furthermore, a T cell-restricted *Srsf1*-deficient mice develops systemic autoimmunity and lupus-nephritis (*Katsuyama et al., 2019*; *Paz et al., 2021*), whereas *Srsf1* deletion in myogenic progenitors leads to defects in neuromuscular junctions (*Liu et al., 2020*).

This evidence highlights the splicing roles of SRSF1 in cellular transformation and tissue development; however, dissecting splicing from post-splicing activities of SRSF1 cannot be resolved through such SRSF1 depletion studies. As such, the physiological relevance of its post-splicing activities has remained fully enigmatic and alternate separation-of-function strategies are necessary to determine to what extent SRSF1 might regulate mRNA translation in vivo.

Here, we have engineered a mouse model expressing an exclusively non-shuttling endogenous SRSF1 protein retained in the nucleus. We show that post-splicing activities of SRSF1 in the cytoplasm are dispensable for embryonic development; however, mutant animals display multiple traits associated with dysfunctional motile cilia and sperm flagella. Accordingly, we also observed that the lack of cytoplasmic SRSF1 leads to reduced translation of ciliary mRNAs in testes, accompanied by decreased abundance of the encoded proteins both in tracheal cultures and testes. These proteins, previously associated with traits comparable to those observed affected in $Srsf1^{NRS/NRS}$ animals, are part of the multiciliogenesis program that regulates cilia and flagellar movement. We conclude that nucleo-cytoplasmic shuttling of SRSF1 is indeed required for proper development and primarily affects mRNA translation, contributing to the biogenesis and function of motile cilia.

## Results

### Generation of a non-shuttling SRSF1 protein mouse model

To create an in vivo mouse model expressing only a nuclear-retained SRSF1 protein, we inserted a potent nuclear retention signal (NRS) at the C-terminus of the *Srsf1* genomic locus (*Figure 1A*). This sequence is naturally present in the non-shuttling SR protein SRSF2 and when fused to SRSF1 prevents its shuttling when overexpressed (*Cazalla et al., 2002*; *Maslon et al., 2014*; *Sanford et al., 2004*). We designed a CRISPR/Cas9-assisted strategy in which an NRS sequence, a small linker and a T7 tag were introduced at the C-terminus of the canonical SRSF1 isoform (*Sun et al., 2010*; *Figure 1B*, *Figure 1—figure supplement 1*). This approach minimizes perturbation of the

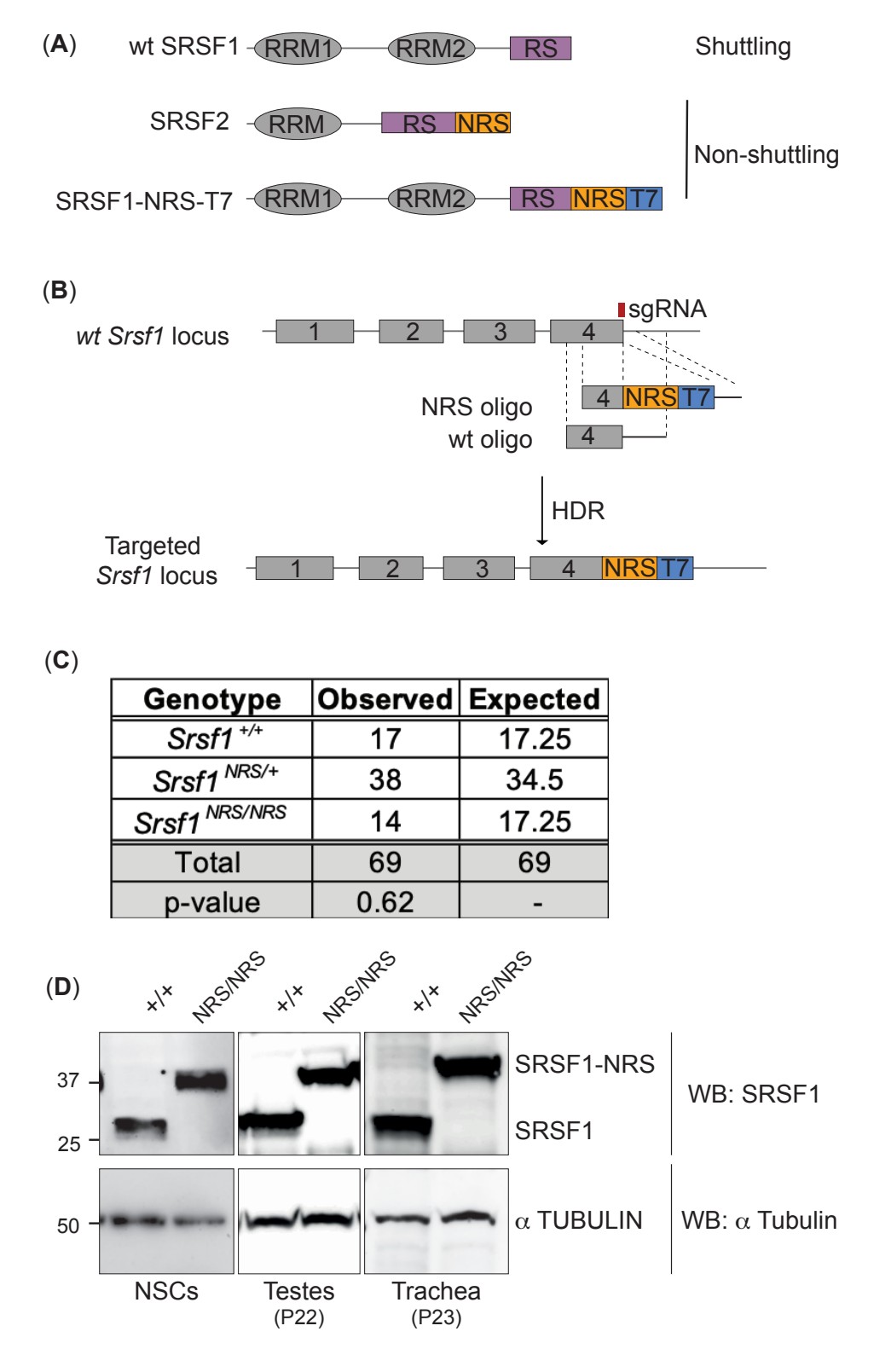

**Figure 1.** Generation of a nuclear-restrained SRSF1 ($Srsf1^{NRS/NRS}$) knock-in mouse model. (A) Domain structure and nucleo-cytoplasmic shuttling ability of wild-type (WT) SRSF1 and SRSF2 proteins, and the artificial fusion protein SRSF1-NRS. (B) Schematic representation of the $Srsf1$ locus and the CRISPR/Cas9 strategy used to introduce an NRS-T7 sequence at its C-terminus. The nucleotide sequence of the introduced nuclear retention signal (NRS) is identical to that present in the endogenous mouse $Srsf2$ gene (*Cazalla et al., 2002*). (C) $Srsf1^{NRS/NRS}$ homozygous knock-in mice

*Figure 1 continued on next page*

*Figure 1 continued*

complete embryogenesis and are viable postnatally. The number of pups obtained from *Srsf1*$^{+/NRS}$ intercrosses with indicated genotypes at postnatal day 14 are indicated. The expected Mendelian numbers and $\chi^2$-test p value (p=0.62, n = 69) are also shown. (D) Expression of SRSF1 and SRSF1-NRS proteins in neuronal stem cells (NSCs), testes, and trachea derived from *Srsf1*$^{NRS/NRS}$ mice at the indicated postnatal age. An anti-SRSF1 antibody was used for western blot analysis to detect both the endogenous SRSF1 protein, as well as the knock-in protein (SRSF1-NRS), whereas anti-tubulin was used as a loading control.

The online version of this article includes the following source data and figure supplement(s) for figure 1:

**Source data 1.** Original western blot images.
**Figure supplement 1.** Genotyping of *Srsf1*$^{NRS/NRS}$ mice.
**Figure supplement 1—source data 1.** Original western blot images.
**Figure supplement 2.** Expression of SRSF1 and SRSF1-NRS protein in cell lines and tissues derived from *Srsf1*$^{+/+}$ *and Srsf1*$^{NRS/NRS}$ mice.
**Figure supplement 3.** Localization of SRSF1 and SRSF1-NRS protein.

N-terminal RRM domains of SRSF1 that are crucial for RNA recognition and its function in splicing (*Clery et al., 2013*) The presence of the T7 tag facilitates both visualization of the tagged protein (referred herein as SRSF1-NRS), as well as biochemical experiments. In contrast to the completely penetrant early embryonic lethality of *Srsf1*$^{-/-}$ knockout mice (*Xu et al., 2005*), *Srsf1*$^{NRS/NRS}$ mice were born from *Srsf1*$^{+/NRS}$ intercrosses and survived postnatally to the point of genotyping (day 14) (*Figure 1C*). However, postnatal *Srsf1*$^{NRS/NRS}$ mice displayed overt phenotypic abnormalities from this stage onwards (see later). These findings indicate that the cytoplasmic functions for SRSF1 allele are mostly dispensable for embryonic development. We confirmed that the SRSF1-NRS fusion protein is expressed in in all tissues and cell lines tested derived from homozygous *Srsf1*$^{NRS/NRS}$ mice (*Figure 1D, Figure 1—figure supplement 2*).

We and others have previously shown that the SRSF1-NRS chimeric protein is retained in the nucleus (*Cazalla et al., 2002*; *Lin et al., 2005*). Here, we analyzed the shuttling ability of endogenous SRSF1-NRS protein in neural stem cells (NSCs) differentiated in vitro from embryonic stem cells (ESCs) derived from *Srsf1*$^{NRS/NRS}$ animals using an inter-species heterokaryon assay (*Piñol-Roma and Dreyfuss, 1992*). As expected, we observed that SRSF1-NRS protein was only detected in the mouse but not in human nuclei, confirming that it is indeed restricted to the mouse nucleus (*Figure 1—figure supplement 3*, lower panel). As a positive control, we transiently expressed a T7-tagged WT-SRSF1 in mouse NSCs and clearly observed its presence in the recipient HeLa nuclei, underlining the innate shuttling activity of WT SRSF1 (*Figure 1—figure supplement 3*, upper panel). This confirms that our targeting strategy was successful and the resulting SRSF1-NRS fusion protein expressed from the endogenous *Srsf1* locus is in fact restricted to the nucleus.

## Srsf1$^{NRS/NRS}$ mice are growth restricted

The viability of *Srsf1*$^{NRS/NRS}$ embryos contrasts with the reported lethality of *Srsf1* null embryos by E7.5 (*Xu et al., 2005*). Strikingly, homozygous *Srsf1*$^{NRS/NRS}$ mice displayed numerous severe postnatal phenotypes. First, these knock-in mice were visibly smaller in size, including those that survived up to 7.5 months of age, being on average 30% lighter in weight than littermate controls (Student's t-test, p<0.0001), suggestive of a growth restriction phenotype (*Figure 2A*).

To investigate whether the growth restriction observed in *Srsf1*$^{NRS/NRS}$ mice arises during embryogenesis, E12.5 embryos and placentas were harvested from four *Srsf1*$^{+/NRS}$ intercross litters. *Srsf1*$^{NRS/NRS}$ embryos were grossly phenotypically normal at this stage of development (*Figure 2B*), with no difference in embryo or placenta weights compared to their *Srsf1*$^{+/+}$ (*Figure 2C*). These results indicate that the observed growth restriction does not arise from impaired embryogenesis or abnormal maternal nutrient transfer across the placenta, suggesting that postnatal growth is specifically affected in *Srsf1*$^{NRS/NRS}$ animals.

## *Srsf1*$^{NRS/NRS}$ mice perinatal phenotypes are indicative of defects in motile cilia

In addition to restricted growth, half of the *Srsf1*$^{NRS/NRS}$ animals developed hydrocephalus by P14. In contrast, no cases of hydrocephalus were observed in either the *Srsf1*$^{+/+}$ or *Srsf1*$^{NRS/+}$ littermates (*Figure 3A*). Hydrocephalus is caused by the accumulation of cerebrospinal fluid (CSF) resulting in dilatation of brain ventricles and increased pressure on the surrounding brain tissue, which can lead

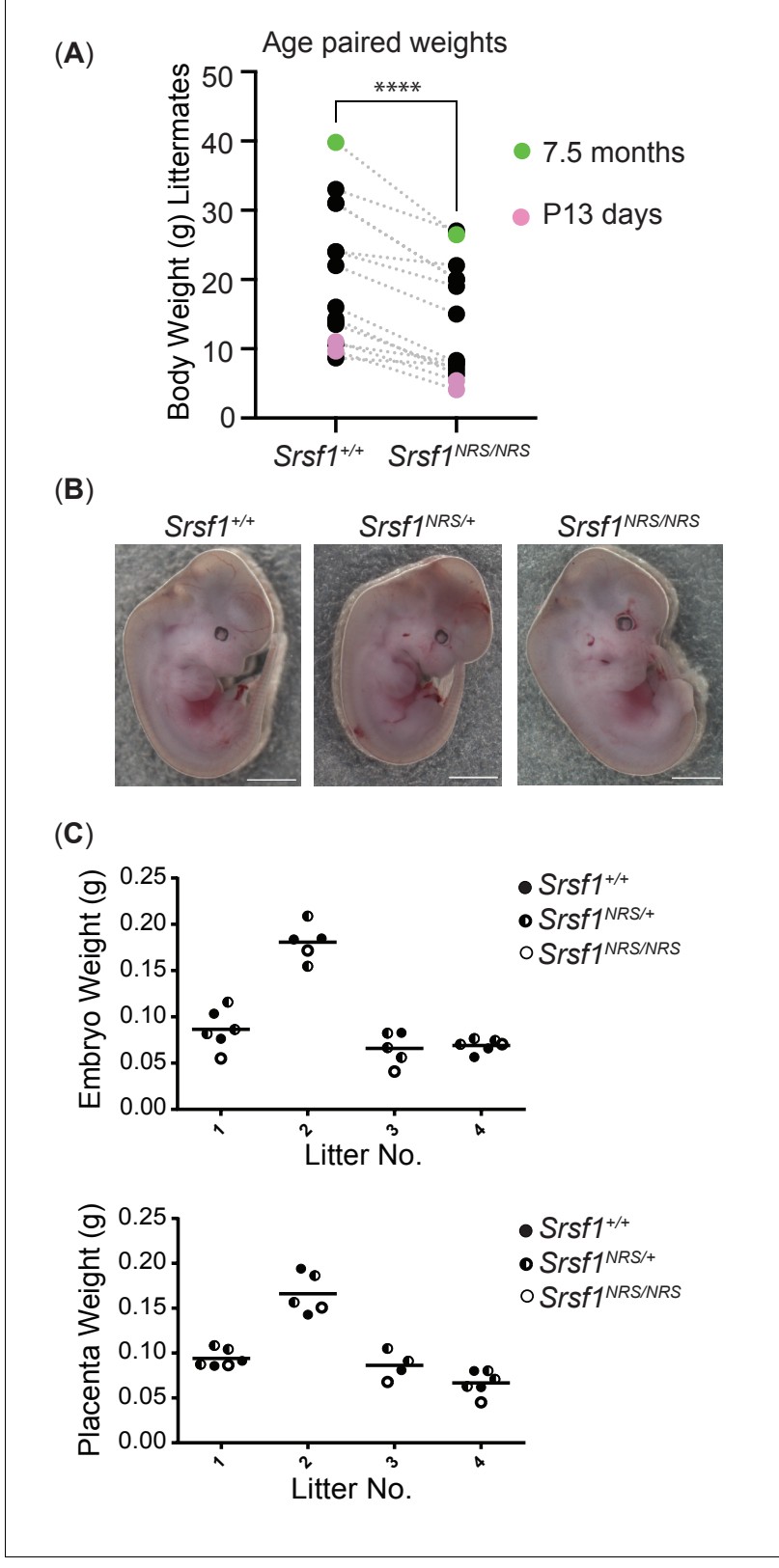

**Figure 2.** Homozygosity for the *Srsf1$^{NRS}$* allele causes postnatal growth restriction. (**A**) Whole-body weight of sex-matched littermate pairs indicated by connecting lines. Animals range from 13 days (pink) to 7.5 months old (blue). N = 15 animals/genotype. (**B**) *Srsf1$^{NRS/NRS}$* knock-in embryos at E13.5 are grossly phenotypically normal. *Srsf1$^{NRS/NRS}$* embryos were represented in these litters at the expected Mendelian ratio (7 *Srsf1$^{+/+}$*, 11 *Srsf1$^{+/NRS}$*, 4

*Figure 2 continued on next page*

*Figure 2 continued*

*Srsf1$^{NRS/NRS}$*; $\chi^2$p value = 0.66). (C) Scatter plots showing the weight of whole embryos (top panel) and of placentas from four independent litters of E12.5 embryos (bottom panel).

to cranial doming during the perinatal period (*McAllister, 2012*). As in humans, hydrocephalus can result in progressive brain enlargement, neurological dysfunction, seizures, and death. Hydrocephalus can be caused by obstruction of the aqueducts or abnormal beating of cilia lining the brain ventricles, thus preventing CSF flow (*Ibañez-Tallon et al., 2004*). Although coronal sections of P14 brains showed dilation of brain ventricles and damage to surrounding neurons, we found no evidence of aqueduct obstruction even in most severe cases of hydrocephalus (*Figure 3B*). *Srsf1$^{NRS/+}$* animals were crossed onto a cilia cell cycle biosensor *ARL13B-Fucci2A* transgenic mice (*Ford et al., 2018*). Here, ubiquitous expression of the ciliary reporter ARL13B-Cerulean allows for live imaging of all ciliary types and revealed that ependymal and choroid plexus cilia are present in *Srsf1$^{NRS/NRS}$* animals (*Figure 3—figure supplement 1*).

In addition to hydrocephalus and growth restriction, *Srsf1$^{NRS/NRS}$* males presented sperm with abnormal head morphology and a large proportion of immotile flagella (*Figure 3C*), whereas those remaining motile exhibited abnormal waveforms (*Figure 3D*). Electron microscopy analysis also revealed ultrastructural defects in the motile ciliary axonemes of mouse tracheal epithelial cultures (mTECs) from *Srsf1$^{NRS/NRS}$* mice, including lack of central pair complex (1/81), supernumerary central microtubules (10/81), or single outer microtubules (2/81) compared to control *Srsf1$^{+/+}$* cultures (one single outer microtubule out of n = 94) (Figure 5G; $X^2$ of all microtubule defects < 0.001) (*Figure 3E*). Together these results suggest that defect in the structure and/or function of motile cilia may underlie multiple postnatal phenotypes observed in *Srsf1$^{NRS/NRS}$* animals including growth arrest, hydrocephaly, and reduced fertility.

## Molecular signatures of compromised cilia and flagellar motility in *Srsf1$^{NRS/NRS}$* animals

In order to investigate the mechanisms by which cilia motility is perturbed in *Srsf1$^{NRS/NRS}$* animals, we used synchronized primary mTECs derived from sets of *Srsf1$^{NRS/NRS}$* and *Srsf1$^{+/+}$* littermates to allow the stepwise study of the de novo production of multiciliated cells (*Vladar and Brody, 2013*), including number of cilia formed by centriole amplification as well as changes in cilia content (*Oltean et al., 2018*). From progenitors bearing a single non-motile primary cilia to fully differentiated epithelial cells bearing hundreds of motile cilia, these cultures faithfully replicate airway developmental programs (*Jain et al., 2010*) during the course of a few weeks (*Figure 4A*). Motile cilia move randomly from the onset after the rapid burst of cytoplasmic synthesis, assembly, and transport of motile-specific components into apically docked cilia primordia, but their waveform changes as cilia mature and movement between ciliary bundles becomes coordinated propelling mucus across the surface of these multiciliated epithelia (*Figure 4—video 1*). The complexity and gradual waveform changes depend on multiple components not limited to ciliary motors (inner dynein arm [IDA] and outer dynein arms [ODA]) that power ciliary beating, but also include structural support of motile axonemes (Tektins and Nexin-Dynein Regulatory complexes [N-DRC]) (*Figure 4B*). Mutations in such components result in congenital dysfunction of motile cilia, termed primary ciliary dyskinesia (PCD) (*Loges et al., 2009*; *Mitchison et al., 2012*; *Tanaka et al., 2004*; *Wirschell et al., 2013*). High-speed video microscopy and immunofluorescence of mTEC cultures during differentiation indicate that although the production of multiciliated cells is not affected (*Figure 4—figure supplement 1A*), a significant proportion of cilia remain immotile in *Srsf1$^{NRS/NRS}$* inserts until ALI13 (*Figure 4C*). Moreover, even at these later stages, mutant motile ciliary bundles exhibit altered beat patterns in relation to the control cultures, as illustrated by kymographs, ciliary beat frequencies (*Figure 4D, E*), and beat coordination between ciliary bundles (*Figure 4—figure supplement 1B*). The motility defects observed in tracheal cilia are consistent with those observed in sperm flagella (*Figure 3D*), suggesting that shared molecular mechanisms underlie these abnormal traits.

To investigate the molecular changes underlying these defects, we first performed unsupervised analyses of total proteomes from day 4 post-airlift (ALI4) to day 18 (ALI18) using the subset of proteins, which have been annotated with the GO term 'cilia.' We found some cilia candidates that

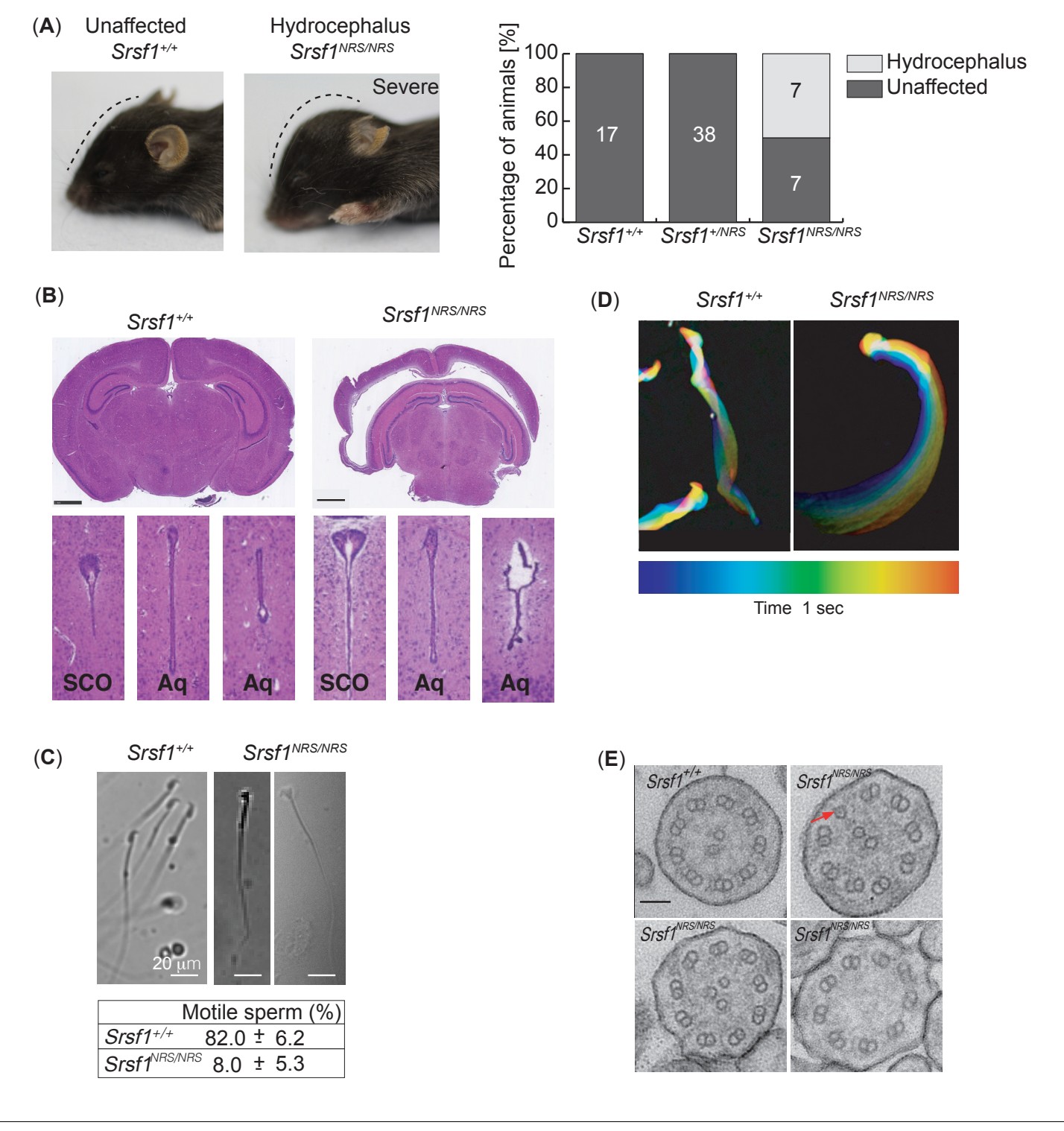

**Figure 3.** Homozygous *Srsf1*<sup>NRS/NRS</sup> mice develop hydrocephalus. (**A**) *Srsf1*<sup>NRS/NRS</sup> mice show signs of developing hydrocephalus (the curvature of the skull is depicted by a dashed line). Half of the mice culled from the first two cohorts developed externally visible hydrocephalus of a varying severity by P14, while hydrocephalus was not observed in *Srsf1*<sup>+/+</sup> or *Srsf1*<sup>NRS/+</sup> littermates. The bar plot indicates the incidence of mice unaffected or with gross hydrocephalus (percentage and total number) in the two cohorts of mice used (p-value=0.0013 and <0.0001, respectively; Fischer's exact test). (**B**) H& E staining of coronal sections of brains at subcommissural organ (SCO) and aqueducts (Aq) of third and fourth ventricles show no overt stenosis. Scale bar: 1 mm. (**C**) Representative images illustrate abnormal head shape of some spermatozoa observed in *Srsf1*<sup>NRS/NRS</sup> littermates. Bottom panel shows percentage of motile sperm (N ≥ 3 animals per genotype). (**D**) Color coding illustrates that the complex rotational pattern of *Srsf1*<sup>+/+</sup> spermatozoa,

*Figure 3 continued on next page*

Figure 3 continued

required to propel the sperm forward, is absent in the few motile *Srsf1^NRS/NRS^* spermatozoa. (E) Transmission electron microscopy of transverse sections of mouse tracheal epithelial culture (mTEC) cilia showing (9+2) microtubules in *Srsf1^+/+^* or abnormal variations found in *Srsf1^NRS/NRS^*. Red arrow illustrates microtubule singlet. Scale bar: 100 nm.

The online version of this article includes the following figure supplement(s) for figure 3:

**Figure supplement 1.** Live images of ependymal cells and choroid plexus from *R26Arl13b-Fucci2aR^Tg/Tg^*,*Srsf1^+/+^*and *R26Arl13b-Fucci2aR^Tg/Tg^*,*Srsf1^NRS/NRS^* (P27) mice.

were concurrently and significantly upregulated at later stages of differentiation in *Srsf1^+/+^* controls that failed to be induced in a coordinated manner in *Srsf1^NRS/NRS^* cultures (*Figure 4F*). The major changes between *Srsf1^NRS/NRS^* and *Srsf1^+/+^* proteomes involved downregulation of proteins with GO terms related to 'assembly of motile cilia' (*Figure 4—figure supplement 1C, D*). Interestingly, the affected proteins include dynein motor assembly factors (DNAAFs) and axonemal dynein motors (DNAHs), as well as Tektins and N-DRC components that are required for normal axoneme assembly and proper motility of mature ciliary bundles (*Figure 4G*). The abnormal levels of motile cilia components found in *Srsf1^NRS/NRS^* proteomes are consistent with the observed ciliary dysfunctions and structural defects observed by TEM (*Figure 3E*).

Moreover, these results were confirmed by performing total testes proteome analysis at P22–23, a stage with synchronized spermiogenesis and flagellar extension, which would correspond with elevated cytoplasmic synthesis and pre-assembly of flagellar precursors (*Mali et al., 2018*).

Indeed, a significant decrease of the same functional groups observed in tracheal proteomes was seen in Srsf1^NRS/NRS^ testes (*Figure 4H*), broadly, motor assembly factors (DNAAFs), axoneme bending regulators (DRCs), and support protein (CFAPs, RSPHs, and Tektins) without broader developmental delays (*Figure 4—figure supplement 1*). Together these results support a general requirement for cytoplasmic SRSF1 during motile ciliogenesis.

## The SRSF1-NRS protein does not grossly affect mRNA splicing or global mRNA export

Phenotypes observed in the *Srsf1^NRS/NRS^* mice provide evidence that the lack of cytoplasmic SRSF1 perturbed developmental pathways involved in motile cilia. SRSF1 is essential for cellular and organism viability, which has been attributed to its splicing role in the nucleus (*Lin et al., 2005*; *Xu et al., 2005*). The viability of *Srsf1^NRS/NRS^* knock-in mice strongly suggests that SRSF1-mediated splicing was not grossly affected in the homozygous animals. In addition, previous work using an exogenous SRSF1-NRS protein revealed that the nuclear splicing role of SRSF1 was essential and could be decoupled from its ability to shuttle (*Lin et al., 2005*). We performed deep RNA-sequencing analysis on mouse embryonic fibroblasts (MEF) derived from *Srsf1^NRS/NRS^*, *Srsf1^NRS/+^*, and *Srsf1^+/+^* littermates (*Figure 5—figure supplement 1A* and *Figure 5—source data 1*). Splicing changes were analyzed using SUPPA2, a tool which displays differential splicing between conditions as changes in proportion spliced-in (ΔPSI) (*Trincado et al., 2018*). As such, SUPPA2 is advantageous as it permits analysis of multiple samples and accounts for biological variability between samples. This was important as our biological replicates were isolated from different embryos with the same genotype, which will have a certain degree of intrinsic heterogeneity. Pairwise comparisons of *Srsf1^+/+^*, *Srsf1^NRS/+^*, and *Srsf1^NRS/NRS^* MEFs demonstrated that there were no gross splicing changes induced by the presence of the NRS insertion (*Figure 5—figure supplement 1A* and *Figure 5—source data 1*). Out of 65, 318 transcripts analyzed, we observed only 26 changes in splicing considering ΔPSI => 0.2 (p <= 0.01), which is comparable to the number of changes between *Srsf1^NRS/+^* and *Srsf1^+/+^* MEFs. Considering that *Srsf1^NRS/+^* mice do not have a quantifiable phenotype, the changes we observe likely represent stochastic changes in AS that could be explained by environmental or sampling factors. This confirms that precluding the shuttling ability of SRSF1 preserves its splicing function.

Shuttling SR proteins, including SRSF1, have been also proposed to function in nuclear mRNA export, through interactions with the export factor TAP/NXF1 (*Hargous et al., 2006*; *Huang et al., 2003*; *Huang and Steitz, 2001*). We performed high-throughput RNA sequencing analysis on cytoplasmic fractions from MEFs derived from *Srsf1^+/+^* or *Srsf1^NRS/NRS^* animals or NSCs differentiated in vitro from ESCs derived from *Srsf1^+/+^* or *Srsf1^NRS/NRS^* animals (*Figure 5—figure supplement 1B* and

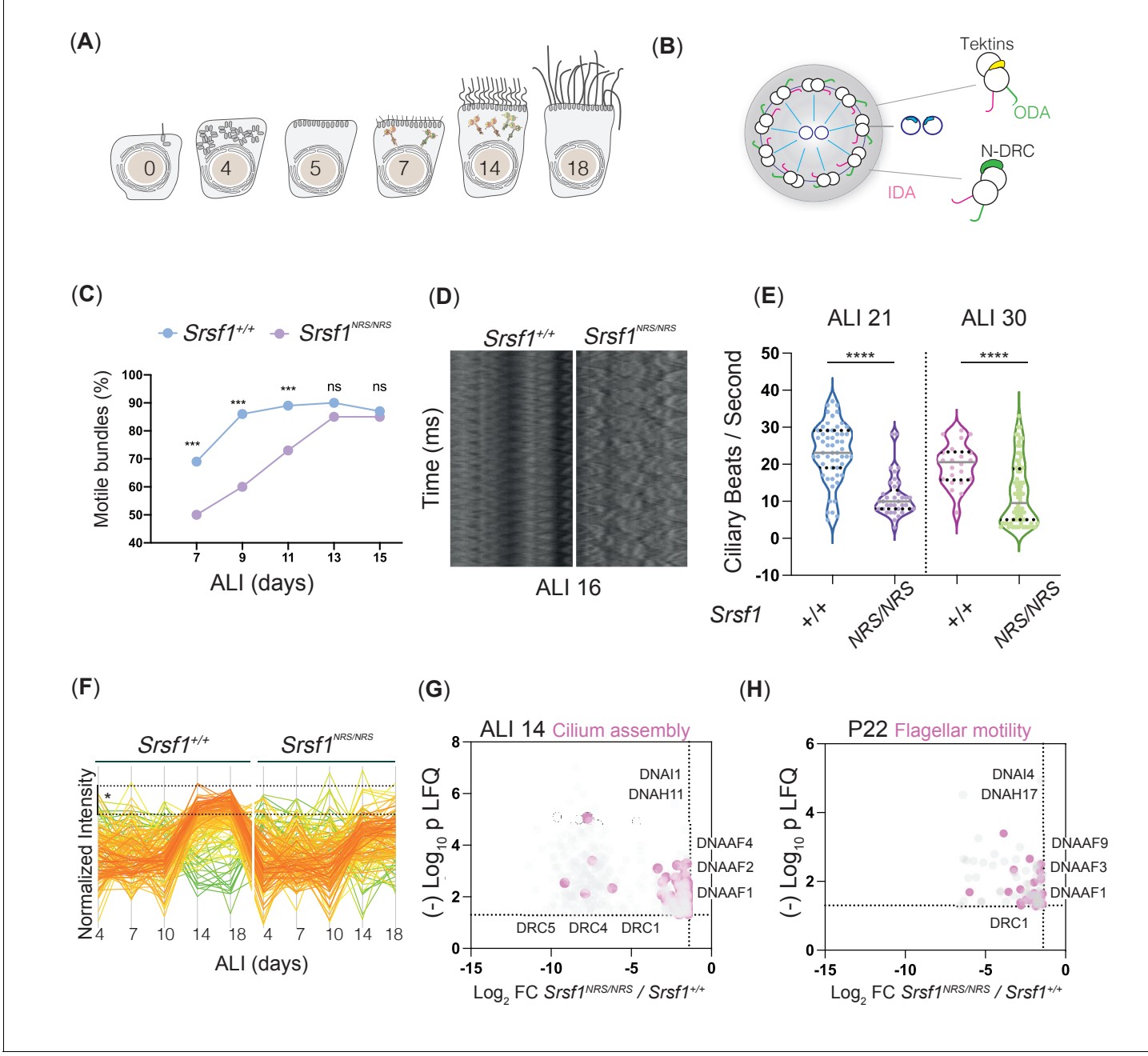

**Figure 4.** Molecular studies of *Srsf1*^*NRS/NRS*^ mice are consistent with defects in ciliary motility. (A) Diagram depicts landmark events in the maturation of motile cilia in mouse tracheal epithelial cultures (mTECs), upon culture in air-liquid interface (ALI) as days post airlift. Stages depicted include centriole amplification and apical docking (ALI 4–6), burst of synthesis, and assembly of motile ciliary machinery concomitant with growth and ciliary elongation (ALI 7–14) continuous ciliary beat maturation (ALI 18). (B) Diagram of a motile axoneme and some auxiliary components . N-DRC: nexin-dynein regulatory complexes; ODA: outer dynein arm; IDA: inner dynein arm. (C) Percentage of motile ciliary bundles during mTEC differentiation. N = 2 animals/genotype at ALI7 and ALI9; N = 3 at all later stages. More than >100 ciliary bundles were scored at each time point exact test at each stage. (D) Representative kymographs of motile bundles illustrate differences in beating amplitude at stages late stages, ALI 16. (E) Cilia beat frequency of *Srsf1*^*NRS/NRS*^ and *Srsf1*^*+/+*^ tracheal cultures grown in parallel at the indicated differentiation stages. Asterisks denote p<0.001 determined by Mann–Whitney tests as *Srsf1*^*NRS/NRS*^ ciliary movement do not follow a normal distribution. (F) Z-normalized intensities of proteins containing 'cilia' within their GO term aligned along tracheal stages (ALI4–18) and genotypes. Each line represents a single protein, where color coding denotes those that match closely to the mean trajectory of the group (red) from those that deviate (green). This shows that *Srsf1*^*+/+*^ induces the coordinated expression of multiple cilia-associated proteins during maturation with a greater amplitude than *Srsf1*^*NRS/NRS*^. Note the tighter distribution of trajectories, and greater fold change (FC) in *Srsf1*^*+/+*^ samples (*). (G, H) Volcano plots of proteins that are significantly under-represented in *Srsf1*^*NRS/NRS*^ cultures at ALI14 (G) and

*Figure 4 continued on next page*

*Figure 4 continued*

testes at P22 (H). Pink dots represent proteins with cilia in their GO term. Label-free quantitation (LFQ) values, analysis, and protein identity of data presented in panels (G) and (H) can be found in *Figure 4—source data 1*.

The online version of this article includes the following video, source data, and figure supplement(s) for figure 4:

**Source data 1.** Identity and intensity values of proteins proteins underrepresented in Srsf1NRS/NRS mTEC and testes total proteomes.

**Figure supplement 1.** Nuclear sequestration of SRSF1 leads to alterations in total proteomes consistent with defects in cilia motility observed during mouse tracheal epithelial culture (mTEC) differentiation.

**Figure supplement 1—source data 1.** Identity and intensity values of all proteins identified in mTEC and testes total proteomes.

**Figure 4—video 1.** Composite image illustrating how ciliary movement within and between multiciliated cells changes as mouse tracheal epithelial cultures mature in ALI cultures.

https://elifesciences.org/articles/65104#fig4video1

*Figure 5—source data 2*). We found that the abundance of 121 and 57 mRNAs was decreased in the cytoplasm of *Srsf1^{NRS/NRS}* MEFs and NSCs, respectively. In previous work, 225 transcripts were identified as the direct export targets of SRSF1 in P19 cells (*Müller-McNicoll et al., 2016*); however, we found no overlap between those targets and mRNAs that change in this study. Similarly, cell fractionation followed by RT-qPCR of selected mRNAs in *Srsf1^{+/+}* or *Srsf1^{NRS/NRS}* cultures revealed that the NRS insertion affected cytoplasmic levels of very few export candidates tested (*Figure 5—figure supplement 1C*). Taken together, our data shows that a nuclear-restricted SRSF1 does not compromise pre-mRNA splicing and is associated with relatively few changes in mRNA export.

## Reduced translation of a subset of mRNAs in Srsf1^{NRS/NRS} NSCs

To determine whether the presence of a nuclear-retained SRSF1 protein affects mRNA translation in vivo, we performed polysomal shift analyses in different cell lines derived from *Srsf1^{+/+}* and *Srsf1^{NRS/NRS}* animals. The shuttling ability of SRSF1 is not required for embryonic development but becomes important in later postnatal stages (*Figures 2* and *3*), suggesting that the shuttling ability of SRSF1 becomes more important as cells differentiate. Therefore, we selected ESCs as an undifferentiated pluripotent cell type, NSCs to explore changes upon differentiation into lineage-restricted cell types relevant to the observed phenotypes of *Srsf1^{NRS/NRS}* animals, and MEFs as a terminally differentiated cell type. Cytoplasmic RNA was harvested from three biological replicates and RNA-seq, and polysomal profiling was carried out to measure the transcriptome and translatome, respectively (*Figure 5A* and *Figure 5—source data 3*). We confirmed the expression of the pluripotency gene *Oct4* (*Pou5f1*) in ESCs, whereas the co-expression of the NSC markers *Nestin* (*Nes*) and *Sox2 in NSCs* indicated that the neural lineage has been successfully induced (*Figure 5—source data 2*). Finally, *Thy1* was chosen as a fibroblast marker for MEF cultures. Cytoplasmic extracts were fractionated across 10–45% sucrose gradients and RNA isolated from subpolysomal and heavy polysomal fractions, followed by high-throughput sequencing, as previously described (*Maslon et al., 2014*).

To accurately identify those mRNAs whose translation is responsive to the presence of cytoplasmic SRSF1 and remain associated with ribosomes during the fractionation procedure, we first calculated polysome indices (PIs) for each expressed gene, normalized to transcripts per million (TPM) and the polysome shift ratio (PSR) (PSR = log2(PI_SRSF1-NRS/PI_SRSF1)) between *Srsf1^{+/+}* and *Srsf1^{NRS/NRS}* cultures (see Materials and methods). We used these values to determine transcripts that were significantly depleted (p<0.05) from polysomes in *Srsf1^{NRS/NRS}* cells as translation of such transcripts is likely to be directly compromised by the lack of shuttling SRSF1. This analysis revealed substantial changes in the association of mRNAs with polysomes in *Srsf1^{NRS/NRS}* NSCs and MEFs, with a much lower proportion of changes in ESCs (*Figure 5B*). We found that 13 genes (total of 88) in ESCs, 1077 genes (total of 1258) in NSCs, and 464 genes (total of 733) in MEFs were underrepresented in the polysomal fractions of *Srsf1^{NRS/NRS}* samples, strongly suggesting that SRSF1 is directly involved in their translation.

When we compared translational changes considering only genes commonly expressed in all three cell types (9073 in total), we observed that this had marginally affected the number of genes being affected in all three cell types (*Figure 5—figure supplement 2*). Therefore, the observed translational changes are not restricted to lineage-specific transcripts but instead appear to correlate with ongoing differentiation programs or metabolic demands of each cell type. These results

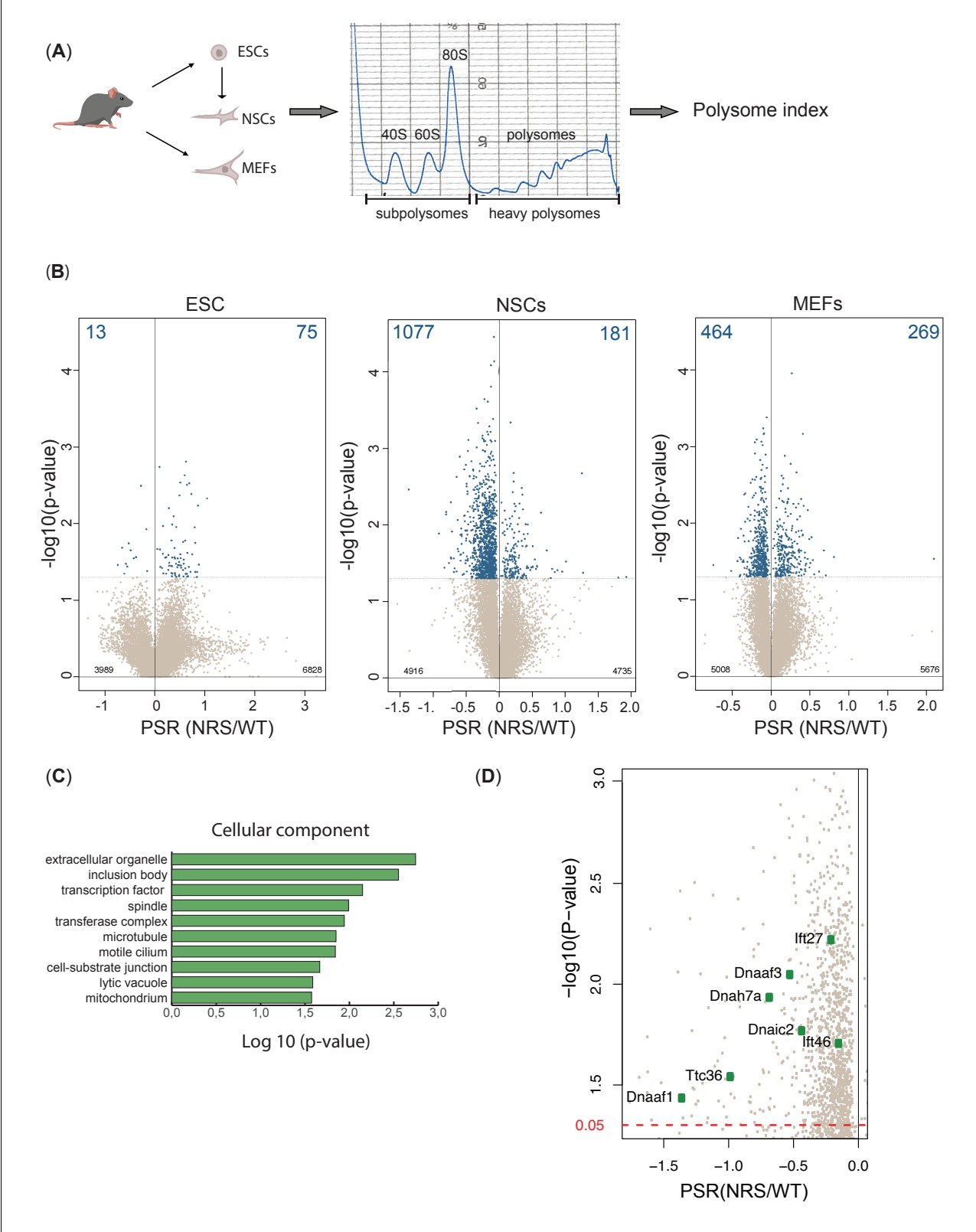

**Figure 5.** Lack of cytoplasmic SRSF1 results in gross changes in translation. (**A**) Schematic of the experimental approach used to identify translation profiles of wild-type SRSF1 or SRSF1-NRS-expressing ESCs, neuronal stem cells (NSCs), and mouse embryonic fibroblast (MEF) cells. A summary of a fractionation profile is depicted. Absorbance at 254 nm was monitored. RNA isolated from the pooled subpolysomal and polysomal fractions was subjected to RNA sequencing. (**B**) Volcano plots showing the distribution of genes expressed in all cell lines according to their polysome shift ratio

*Figure 5 continued on next page*

*Figure 5 continued*

(PSR). PSR is calculated as a ratio of polysome index in *Srsf1*$^{+/+}$ vs. *Srsf1*$^{NRS/NRS}$ cells. Blue dots indicate significant changes. Number of significant changes is indicated in the top corners of each plot. (C) GO term overrepresentation analysis identifies spindle and motile cilium as enriched cellular components category in the downregulated PSR gene list. (D) Plot showing the distribution of genes expressed in NSCs with PSR < 0. Selected ciliary genes are highlighted.

The online version of this article includes the following source data and figure supplement(s) for figure 5:

**Source data 1.** Table 1. Splicing changes between mouse embryonic fibroblast (MEF) lines derived from *Srsf1*$^{NRS/NRS}$ and *Srsf1*$^{+/+}$ littermates.

**Source data 2.** Table 2. RNA sequencing analysis on cytoplasmic fractions from *Srsf1*$^{+/+}$ or *Srsf1*$^{NRS/NRS}$ mouse embryonic fibroblasts (MEFs) and neuronal stem cells (NSCs).

**Source data 3.** Table 3. Polysome shift ratio (PSR) values for genes identified in *Srsf1*$^{+/+}$ or *Srsf1*$^{NRS/NRS}$ ESCs, mouse embryonic fibroblasts (MEFs), and neuronal stem cells (NSCs), respectively.

**Figure supplement 1.** The lack of cytoplasmic SRSF1 does not affect pre-mRNA splicing or mRNA export.

**Figure supplement 2.** Lack of cytoplasmic SRSF1 induces translational changes not restricted to lineage-specific transcript.

**Figure supplement 3.** Polysome shift ratio (PSR) values are independent of alternative splicing and gene expression changes in cytoplasm.

suggest that differentiating cells such as NSCs and to a lesser extent MEFs require SRSF1 for the translation of a subset of mRNAs.

To confirm that the observed changes are indeed translational and not simply due to gene expression changes, we compared differences in AS and cytoplasmic RNA expression levels with the observed translational changes. Specifically, we compared genes with significantly different PSR between *Srsf1*$^{+/+}$ and *Srsf1*$^{NRS/NRS}$ cultures to their respective total mRNA and mRNA isoforms levels from RNA-seq data. Changes in polysome profiling and RNA levels did not correlate in either MEFs or NSCs (***Figure 5—figure supplement 3A***). This indicates that translational changes are a direct response to the loss of SRSF1 in cytoplasm rather than a consequence of altered transcription or mRNA export. Moreover, only a small fraction of differentially expressed splicing isoforms in the cytoplasm (***Figure 5—figure supplement 3B***) were found to be differentially translated (***Figure 5—figure supplement 3C***). Overall, these findings show that the absence of SRSF1 from the cytoplasm has a major impact on translation and that these changes are specific to the cell type even though they primarily affect non-cell-type-specific RNAs. Altogether, this shows that the SRSF1-mediated translation, which was initially observed in vitro and cells in culture, does indeed affect a large subset of mRNAs in the mouse. More importantly, these data also suggest that SRSF1-dependent translation becomes more prominent in differentiating cell types and is consistent with a role of SRSF1 as a translational enhancer during specific cellular demands, as opposed to being a constitutive component of the translational machinery.

Next, we performed Gene Ontology (GO) analysis to determine whether any functional gene category was differentially translated in SRSF1-NRS expressing cells (***Figure 5C***). We considered only transcripts with a change in PSR of >15%. From this analysis, we identified genes associated with spindle category as translationally downregulated in *Srsf1*$^{NRS/NRS}$ NSCs, which agrees with our previous findings in SRSF1-overexpressing 293T cells (***Maslon et al., 2014***) and with the reduced body size of *Srsf1*$^{NRS/NRS}$ animals. Interestingly, the functional category related to motile cilium was also enriched among genes depleted from polysomal fractions of *Srsf1*$^{NRS/NRS}$ NSCs (***Figure 5C***). This is consistent with hydrocephalus, immotile sperm, and abnormally motile cilia in tracheal cells observed in *Srsf1*$^{NRS/NRS}$ animals. In addition, NSCs express a number of mRNAs encoding motile cilia-specific proteins and these were downregulated in *Srsf1*$^{NRS/NRS}$ NSCs polysome fractions (***Figure 5D***). We observed that dynein axonemal assembly factors *Dnaaf1/Lrrc50* and *Dnaaf3*, which are required for the functional assembly of axonemal dynein motors, as well as motor subunits themselves including those of both inner dynein arms (i.e., axonemal dynein heavy chain 7 [*Dnah7a*]) and outer dynein arms (i.e., dynein axonemal intermediate chain 2 [*Dnai2*]) were under-represented in the heavy polysome fractions of *Srsf1*$^{NRS/NRS}$ cultures. Together, these proteins form the functional macromolecular motors that drive ciliary movement, many of which are found mutated in patients with congenital dysfunction of motile cilia, termed primary ciliary dyskinesia (PCD) (***Loges et al., 2009***; ***Loges et al., 2008***; ***Mitchison et al., 2012***; ***Zhang et al., 2002***). Importantly, the abundance of these mRNAs was not altered in the cytoplasm of *Srsf1*$^{NRS/NRS}$ cultures (***Figure 5—source data 2***), which was further confirmed by direct RT-PCR for *Dnaaf1* and *Dnaaf3* (***Figure 5—figure supplement 1C***).

### Reduced translation of a subset of mRNAs in *Srsf1^{NRS/NRS}* testes

Our findings highlight the selective participation of SRSF1 in ongoing differentiation programs and/or metabolic demands of distinct cell types. Since NSC only express a subset of motile ciliary genes (*Mokrý and Karbanová, 2006*), we extended our translational studies to tissues with overt phenotypes and altered proteomes in *Srsf1^{NRS/NRS}* animals. Technical limitations existed in terms of sufficient yields of input material for polysome shifts from mTECs; however, testes provided abundant input material. The observed decreases are in multiple classes of components such as dynein assembly factors (DNAAFs) that assist assembly and trafficking of dynein motor subunits, components of the nexin-dynein regulatory complex (DRCs), which transform microtubule sliding into axonemal bending, the central pair complex and radial spoke proteins (RSPH) that confer structural support, generate waveforms and coordinate motors, as well as structural components like mictrotubule-stabilizing proteins the Tektins to provide mechanical resilience to the fast beating axonemes from tracheal cilia and sperm flagella (*Figure 6A*).

In order to confirm that the observed changes in protein abundance are indeed caused by changes in translational status of encoding mRNAs, we used mouse testes at the same stage of development than those used for proteome analysis and performed polysomal shift coupled to qRT-PCR analysis of select ciliary mRNA transcripts representing each of these functional groups. Strikingly, we found that the polysomal distribution of seven out of eight of these motility-related mRNA transcripts were decreased in *Srsf1^{NRS/NRS}* testes (*Figure 6B, C*). These results demonstrate that mouse testes require SRSF1 for the translation of a subset of mRNAs that encode components of the motile ciliary/flagella machinery. Altogether, this strongly suggests that SRSF1 directly participates in the translation of components of motile cilia and could therefore directly underlie the phenotypes observed in *Srsf1^{NRS/NRS}* pups.

In summary, we show that the *Srsf1^{NRS/NRS}* mouse model mimics organismal and cellular phenotypes attributable to defects in motile cilia, similar to those observed in PCD patients. These phenotypic changes are consistent with a lack of SRSF1-mediated cytoplasmic activities that particularly impact on mRNAs relevant to the motile ciliary machinery, in agreement with altered proteomes and axonemal ultrastructure changes observed in *Srsf1^{NRS/NRS}* animals.

## Discussion

Shuttling SR proteins, including SRSF1, act to coordinate nuclear and cytoplasmic steps of gene expression (*Botti et al., 2017*; *Michlewski et al., 2008*; *Müller-McNicoll et al., 2016*). Indeed, SRSF1 promotes translation of mRNAs encoding RNA processing factors and cell-cycle and centrosome-associated proteins (*Fu et al., 2013*; *Maslon et al., 2014*; *Michlewski et al., 2008*). Despite increasing evidence for post-splicing activities of shuttling SR proteins, their physiological roles in a living organism has remained enigmatic. Here, we set out to comprehensively investigate the in vivo roles and requirement for cytoplasmic SRSF1. We created a mouse model in which SRSF1 is retained in the nucleus by targeting an NRS to SRSF1, resulting in a fusion protein (SRSF1-NRS). In our model, abrogation of cytoplasmic SRSF1 function resulted in several deleterious phenotypes without affecting the nuclear roles of SRSF1. To our knowledge, this is the first demonstration that interfering with the nucleo-cytoplasmic shuttling of an RBP has severe phenotypic consequences giving rise to perinatal phenotypes.

### Multiple postnatal phenotypic changes occur in mice lacking cytoplasmic SRSF1

Contrary to the early embryonic lethality observed with *Srsf1^{-/-}* knockout mice (*Xu et al., 2005*), *Srsf1^{NRS/NRS}* embryos are viable and phenotypically normal (*Figure 2B*), suggesting that the cytoplasmic functions of SRSF1 allele are dispensable for embryonic development. An alternative explanation is that the lack of observed phenotypes in early development could be due to a more permissive shuttling of SRSF2 in pluripotent murine P19 cells, whereas shuttling is abolished upon differentiation (*Botti et al., 2017*). Thus, it remains possible that the chimeric SRSF1-NRS protein, harboring SRSF1 NRS, could potentially display some shuttling during early development. Importantly, we clearly showed that chimeric SRSF1-NRS protein is restricted to the nucleus in NSCs derived from *Srsf1^{NRS/NRS}* mice (*Figure 1—figure supplement 2*).

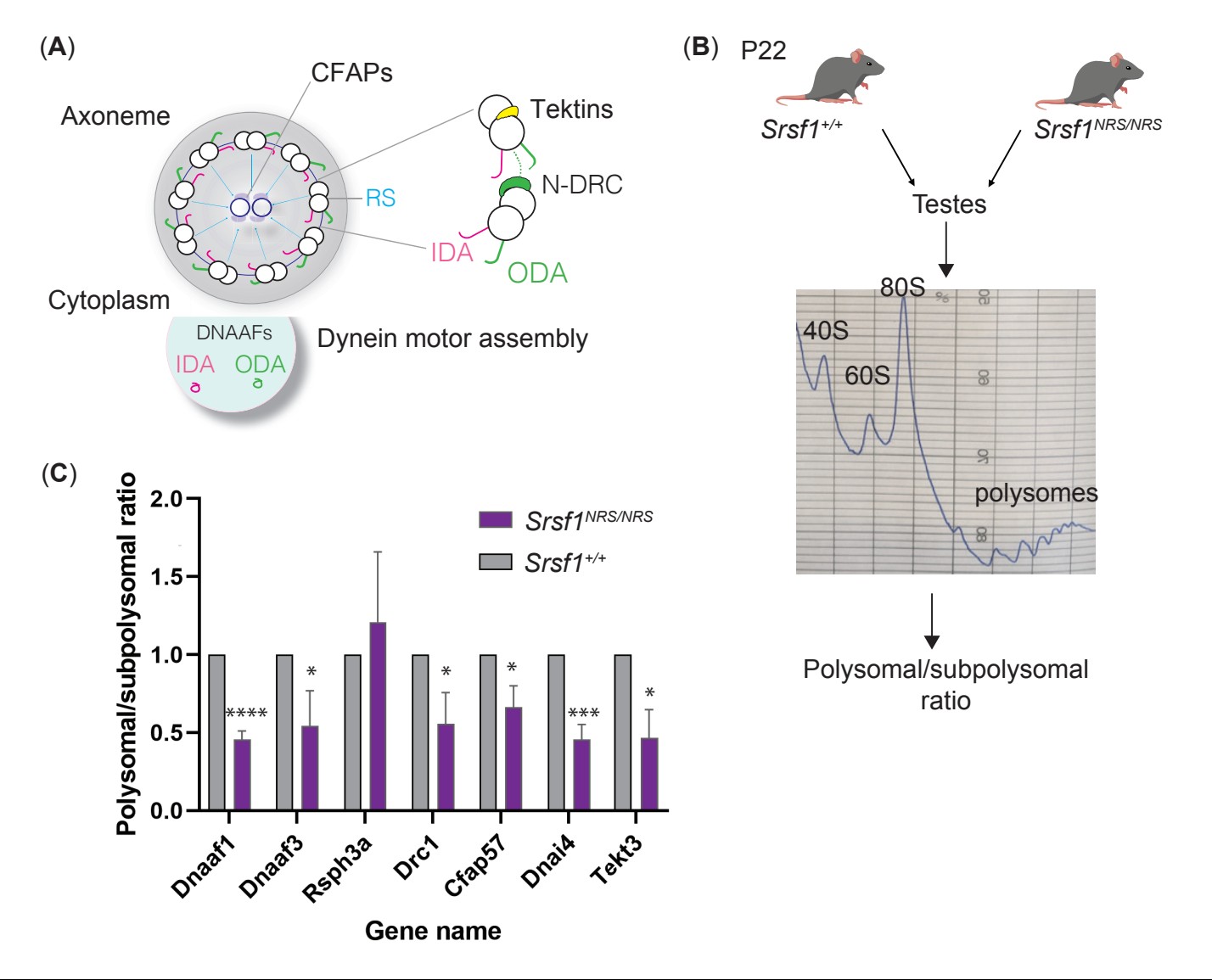

**Figure 6.** Lack of cytoplasmic SRSF1 results in gross changes in translation in mouse testes. (**A**) Summary schematic of functional groups underrepresented in the total proteomes of tracheal cultures and maturing testes. (**B**) Schematic of the experimental approach used to identify translation profiles of *Srsf1*$^{+/+}$ and Srsf1$^{NRS/NRS}$ testes. A summary of a fractionation profile is depicted. Absorbance at 254 nm was monitored. RNA isolated from the pooled subpolysomal and polysomal fractions was subjected to RNA sequencing. (**C**) The polysomal-subpolysomal ratio of individual mRNAs in wild-type or SRSF1-NRS expressing testes was quantified by RT-qPCR. The data is an average of five independent experiments, and each bar represents an average and a standard error. Asterisks represent statistical significance; *p<0.05; ***p<0.0001; ****p<0.00001.

*Srsf1*$^{NRS/NRS}$ mice developed hydrocephalus with variable severity and were smaller than littermate counterparts. Homozygous males also had an increased proportion of immotile sperm, whereas any remaining motile spermatozoa exhibit abnormal motility. Moreover, primary cells from *Srsf1*$^{NRS/NRS}$ airways and spermatozoa displayed reduced cilia beat frequency, disturbed waveforms, and significantly reduced protein levels of motile ciliary components (*Figure 4*). These traits and molecular changes are characteristic of PCD, a genetically heterogenous disease of motile cilia (*Lobo et al., 2015*). Cilia are small microtubule-based projections found on the surface of most mammalian cells types. Defects in cilia function or structure result in a growing spectrum of diverse human diseases known as the ciliopathies (*Reiter and Leroux, 2017*). Whilst we observed no gross phenotypes associated with defects in primary (immotile) cilia in *Srsf1*$^{NRS/NRS}$ animals, multiple organs that depend on motile cilia were affected. Motile cilia are essential for providing coordinated

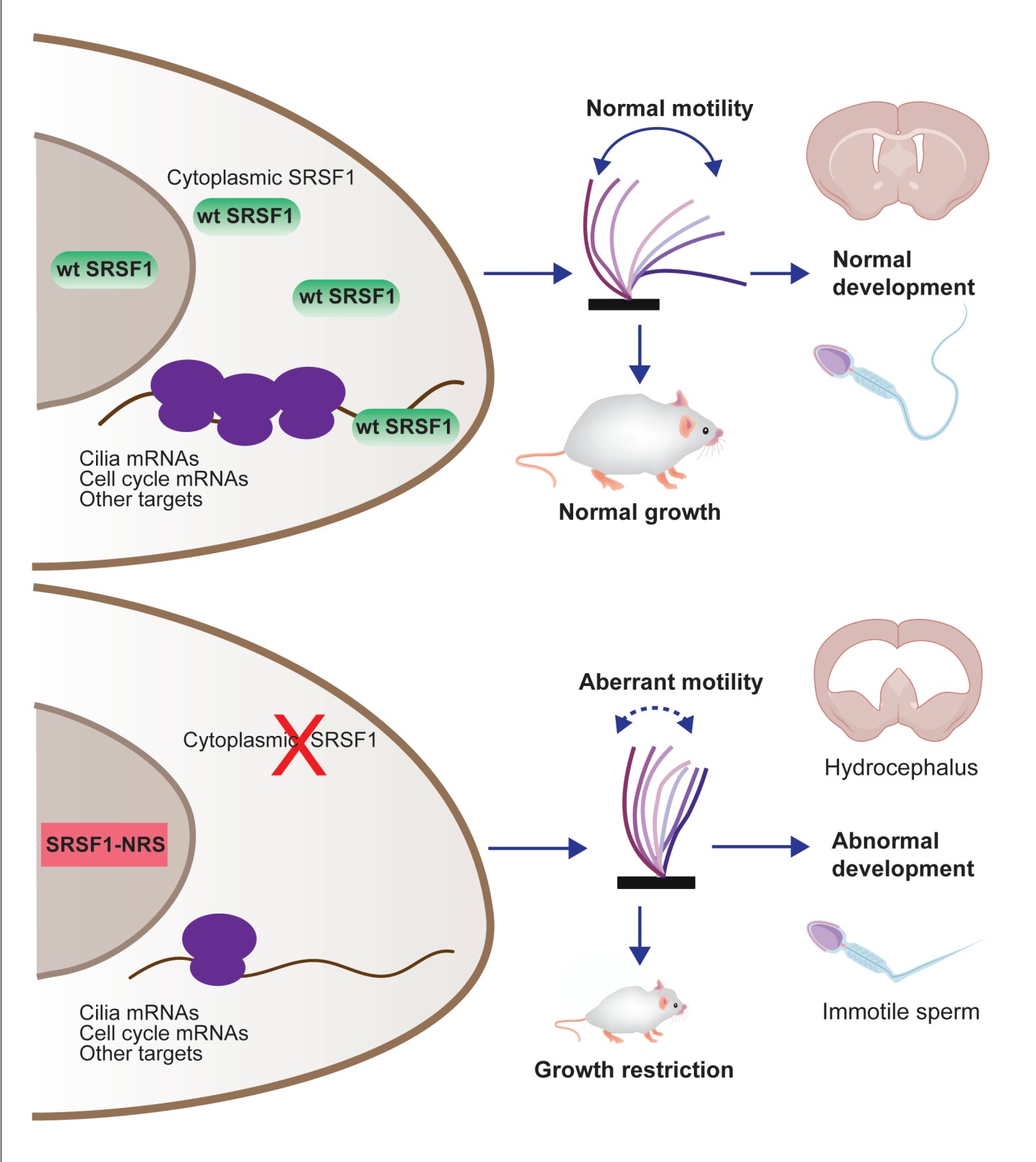

**Figure 7.** Restricting SRSF1 to the nucleus results in perinatal phenotypes in the mouse including growth, brain development, and spermatogenesis, particularly affecting motile cilia. This highlights the physiological role of splicing factor nucleo-cytoplasmic shuttling to reprogram gene expression networks to meet high cellular demands.

mechanical force to drive sperm movement or movement of fluids across tissues affecting brain ventricles, trachea, and spermatozoa. The constellation of phenotypes, abnormal ciliary ultrastructure, altered proteomes, and disrupted motility provide strong evidence that cytoplasmic functions for SRSF1 are essential in vivo during motile cilia differentiation.

To further investigate the effects of depletion of cytoplasmic SRSF1 on developmental proteomes, we used tracheal cultures as a model for the stepwise differentiation of motile ciliogenesis that could contribute to the phenotypes observed in $Srsf1^{NRS/NRS}$ animals. Given the link to SRSF1-dependent translation and mitotic functions at centrosomes in dividing cells (*Maslon et al., 2014*), it was possible that differentiation of multiciliated cells is impacted, given the hundreds of centrioles destined to become the basal bodies of motile cilia need to be generated within a single postmitotic cell (*Meunier and Azimzadeh, 2016*). In particular, depletion of mitotic oscillator component CDK1, previously shown to be a direct target of SRSF1 translation (*Maslon et al., 2014*), hinders differentiation of multiciliated ependymal cells at the stage of massive centriole production (*Al Jord et al., 2017*). However, in the absence of cytoplasmic SRSF1, differentiation appeared to occur on time and give rise to appropriately long multiciliated bundles (*Figure 4—figure supplement 1*). However, cilia motility and movement coordination were specifically compromised, and protein levels of motile ciliary components were significantly reduced (*Figure 4F, G*). Furthermore, motility of the sperm flagella is also affected in $Srsf1^{NRS/NRS}$ animals (*Figure 3C, D*).

During multiciliogenesis, cells transition from bearing a single primary cilium to hundreds of motile cilia requiring de novo synthesis of millions of motor subunits and motile ciliary components in the correct stoichiometry within a short developmental window couple to ciliary elongation. Although a well-defined transcriptional code for induction of cilia motility exists, these transcriptional regulators of multiciliogenesis affect hundreds of motile ciliary genes (*Lewis and Stracker, 2021*) and lead to a drastic decrease in the production of ciliated bundles and cilia number per bundle (*Gomperts et al., 2004*). These traits were not observed in $Srsf1^{NRS/NRS}$ cultures. Recent evidence suggests that spatially regulated translation may be occurring during motile ciliogenesis. During spermatogenesis, *Drosophila* mRNAs encoding motile ciliary proteins are stored in large cytoplasmic granules until their translation is required in the growing sperm axoneme, possibly regulating their translation (*Fingerhut and Yamashita, 2020*). In vertebrates, there is evidence of phase-separated cytoplasmic foci termed dynein axonemal particles (DynAPs) that contain assembly factors, chaperones, and axonemal dynein proteins, which may provide a favorable environment for their orderly assembly and final transport onto cilia (*Huizar et al., 2018*). It remains to be seen whether this is also the case for DynAPs, but as with most other phase-separated condensates, RNA is key to regulate the formation, size, and composition of these phase transitions (*Langdon et al., 2018*; *Maharana et al., 2018*). It is possible that SRSF1 could contribute to localized mRNA delivery and/or translation at these cytoplasmic foci. It could promote the formation of mRNPs that confine such regulated transcripts until a timely co-translational assembly of these large protein complexes is required during motile ciliogenesis.

## Lack of cytoplasmic SRSF1 represses cilia-related mRNA transcripts

We provided evidence showing that abrogation of the cytoplasmic function of SRSF1 led to a decreased polysomal association of a subset of mRNAs in $Srsf1^{NRS/NRS}$ NSCs, yet its impact on AS (nuclear function of SRSF1) or on the abundance of target mRNAs in the cytoplasm (mRNA export) was minimal. This strongly suggests that mRNA translation is specifically debilitated when SRSF1 is absent from the cytoplasm. We found that the lack of cytoplasmic SRSF1 had little effect on translation in ESCs, but it affected translation of over 1000 genes in NSCs. These results are consistent with findings that human ESCs maintain low overall levels of translation, with a global increase in high polysomes upon differentiation towards neural lineages (*Blair et al., 2017*) and thus suggest that SRSF1 participates in developmental programs that rely heavily on translational control. We extended these observations by showing a reduced mRNA translation of a subset of mRNAs with a crucial role in cilia function in $Srsf1^{NRS/NRS}$ testes (*Figure 6*). Thus, the lack of cytoplasmic SRSF1 leads to reduced mRNA translation, with a particular impact on mRNAs related to cilia biogenesis and function of motile cilia broadly impacting development of sperm flagella, as well as in multiciliated tissues such as trachea. There may be an emerging role for RBPs controlling cilia formation and function at the translational level. Intriguingly, CFAP44 (*Coutton et al., 2018*; *Tang et al., 2017*), which is mutated in human male infertility and hydrocephaly, was recently found to be also an RNA-binding protein

localizing to DynAPs (*Drew et al., 2020*), although whether it functions as an RBP in translational control of motile cilia assembly remains unclear. In contrast, Syndesmos (SDOS, NUDT16L1) was shown to bind to and negatively regulate translation of mRNAs responsible for the biogenesis of primary cilia (*Avolio et al., 2018*).

An intriguing possibility is that SRSF1 is able to respond to metabolic demands of the cell needing to reshape its proteome during differentiation. We propose that cytoplasmic activities of SRSF1 are important for the execution of cilia motility programs, providing an extra regulatory level that will ensure responsive and scalable translation at the peak demand of assembly of the complex motile cilia machinery.

In summary, we have developed the first mouse model that allowed us to assess the in vivo relevance of SRSF1 nucleo-cytoplasmic shuttling (*Figure 7*). We conclude that the presence of the splicing factor SRSF1 in the cytoplasm is essential for proper postnatal development, and this is mostly due to the effect of SRSF1 in mRNA translation, which critically affects developmental programs and target mRNAs involved in motile cilia biogenesis and function.

# Materials and methods

## Key resources table

| Reagent type (species) or resource | Designation | Source or reference | Identifiers | Additional information |
|---|---|---|---|---|
| Genetic reagent (*Mus musculus*) | *Srsf1*<sup>NRS/NRS</sup> | This paper | MGI: 645208 | n.a. |
| Cell line (*M. musculus*) | ESCs *Srsf1*<sup>NRS/NRS</sup> | This paper | n.a. | n.a. |
| Cell line (*M. musculus*) | NSCs *Srsf1*<sup>NRS/NRS</sup> | This paper | n.a. | n.a. |
| Cell line (*M. musculus*) | MEFs *Srsf1*<sup>NRS/NRS</sup> | This paper | n.a. | n.a. |
| Antibody | Anti-SRSF1 (mouse monoclonal) | *Hanamura et al., 1998* | | RRID:AB_2533079 |
| Antibody | Anti-T7 (mouse monoclonal) | Novagen | Cat#: 69522 | RRID:AB_11211744 |
| Antibody | Anti-tubulin (YOL1/34) | Abcam | Cat#: ab6161 | RRID:AB_305329 |
| Sequence-based reagent | FAM-5′ Ccdc164 oligonucleotide | IDT | Mm.PT.58.30029646 | |
| Sequence-based reagent | FAM-5′ Tekt3 oligonucleotide | IDT | Mm.PT.58.16533004 | |
| Sequence-based reagent | FAM-5′ Wdr78 oligonucleotide | IDT | Mm.PT.56a.42250015 | |
| Sequence-based reagent | FAM-5′ Wdr65 oligonucleotide | IDT | Mm.PT.58.7105374 | |
| Sequence-based reagent | FAM-5′ Dnaaf1 oligonucleotide | IDT | Mm.PT.58.31362949 | |
| Sequence-based reagent | FAM-5′ Actb oligonucleotide | IDT | Mm.PT.39a.22214843.g | |
| Sequence-based reagent | FAM-5′ Rsph3a oligonucleotide | IDT | Mm.PT.58.29968181.g | |
| Peptide, recombinant protein | Cas9 | TriLink BioTechnologies | L7606 | |
| Peptide, recombinant protein | Q5 High-Fidelity DNA polymerase | New England Biolabs | M0491 | |
| Peptide, recombinant protein | Recombinant Murine EGF | Peprotech | 315-09 | |

*Continued on next page*

*Continued*

| Reagent type (species) or resource | Designation | Source or reference | Identifiers | Additional information |
|---|---|---|---|---|
| Peptide, recombinant protein | Recombinant Human FGF-basic | Peprotech | 100-18B | |
| Peptide, recombinant protein | Laminin | Sigma Aldrich | L2020 | |
| Commercial assay or kit | RNeasy kit | Qiagen | 74106 | |
| Commercial assay or kit | Turbo DNase | Ambion | AM1907 | |
| Commercial assay or kit | SybrGreen RT-QPCR | Roche | 04707516001 | |
| Commercial assay or kit | T7 Quick High Yield RNA Synthesis kit | NEB | E2050S | |
| Chemical compound, drug | Trizol | Life Technologies | 15596026 | |
| Chemical compound, drug | Hoechst 34580 | Sigma Aldrich | 63493 | |
| Chemical compound, drug | Cycloheximide | Sigma Aldrich | C104450 | |
| Chemical compound, drug | PEG1500 | Roche | 10783641001 | |
| Chemical compound, drug | Lipofectamine 3000 | Thermo Fisher Scientific | L3000015 | |
| Chemical compound, drug | MEM non-essential amino acid | Gibco | 11140050 | |
| Chemical compound, drug | Sodium pyruvate | Gibco | 11360070 | |
| Chemical compound, drug | 2-Mercaptoethanol | Gibco | 31350010 | |
| Chemical compound, drug | Accutase | Stemcell | | |
| Chemical compound, drug | BSA (7.5% solution) | Thermo Scientific | 15260037 | |
| Chemical compound, drug | N-2 supplement (100×) | Thermo Scientific | 17502048 | |
| Chemical compound, drug | B-27 Supplement (50×) | Thermo Scientific | 17504044 | |
| Chemical compound, drug | PD0325901 | Stemcell Technologies | 72182 | |
| Chemical compound, drug | CHIR99021 | Miltenyi Biotec | 130-103-926 | |
| Software, algorithm | ImageJ | NIH | https://imagej.nih.gov/ij/ | |
| Software, algorithm | R (v3.5) | N/A | https://www.r-project.org/ | |
| Software, algorithm | STAR | *Dobin et al., 2013* | http://code.google.com/p/rna-star/ | RRID:SCR_004463 |
| Software, algorithm | Salmon | *Patro et al., 2017* | https://combine-lab.github.io/salmon/ | RRID:SCR_017036 |
| Software, algorithm | SUPPA2 | *Trincado et al., 2018* | https://github.com/comprna/SUPPA | |
| Software, algorithm | WebGestalt2019 | *Liao et al., 2019* | http://www.webgestalt.org/ | |
| Software, algorithm | MATS | *Park et al., 2013* | http://rnaseq-mats.sourceforge.net/ | RRID:SCR_013049 |

*Continued on next page*

*Continued*

| Reagent type (species) or resource | Designation | Source or reference | Identifiers | Additional information |
|---|---|---|---|---|
| Software, algorithm | DESeq2 | *Love et al., 2014* | http://www.bioconductor.org/packages/release/bioc/html/DESeq2.html | RRID:SCR_015687 |
| Other | Polysomal profiling RNA-seq data | This paper | GSE161828 | Deposited data |
| Other | MEFs RNA-seq data | This paper | GSE157269 | Deposited data |
| Other | mTECs proteomics data | This paper | PXD019859, username: reviewer79803@ebi.ac.uk password: 4HAr4wOY | Deposited data |

## Animal experiments

We followed international, national, and institutional guidelines for the care and use of animals. Animal experiments were carried out under UK Home Office Project Licenses PPL 60/4424, PB0DC8431, and P18921CDE in facilities at the University of Edinburgh (PEL 60/2605) and were approved by the University of Edinburgh animal welfare and ethical review body.

## CRISPR/Cas9 gene editing in mouse zygotes

The CRISPR target sequence (20-nucleotide sequence followed by a protospacer adjacent motif [PAM] of 'NGG') was selected using the prediction software (http://www.broadinstitute.org/rnai/public/analysis-tools/sgrna-design). Single gRNAs targeting exon 4 of *Srsf1* (Srsf1-205, ENSMUST00000139129.8) were annealed and cloned into the PX458 plasmid (*Cong et al., 2013*). The guide region was then amplified by PCR and paired guide RNAs synthesized by in vitro transcription (T7 Quick High Yield RNA Synthesis kit, NEB, #E2050S). Single-stranded DNA oligonucleotides (WT oligo: 'CGTAGCAGAAGCAACAGCAGGAGTCGCAGTTAC TCCCCAAGGAGAAGCAGAGGATCACCACGCTATTCTCCCCGTCATAGCAGATCTCGCTCTCG TACATAAGATGATTGATGACACTTTTTGTAGAACCCATGTTGTATACAGTTTTCCTTTACTCAG TACAATCTTTTCATTTTTTTAATTCAAGCTGTTTTGTTCAG', NRS oligo: 'AAGCAGAGGATCAC-CACGCTATTCTCCCCGTCATAGCAGATCTCGCTCTCGTACAGGATCCCCTCCGCCCGTG TCGAAGCGAGAGTCCAAGTCTAGGTCGCGGTCCAAGAGCCCACCCAAGTCTCCAGAAGAA-GAGGGAGCAGTTTCTTCCATGGCATCGATGACAGGTGGCCAACAGATGGGTTAAGATGATTGG TGACACTTTTTGTAGAACCCATGTTGTATACAGTTTTCCTTTACTC' and T7 oligo: 'TTAC TCCCCAAGGAGAAGCAGAGGATCACCACGCTATTCTCCCCGTCATAGCAGATCTCGCTCTCG TACAGGATCCCCCGGCGCCGGCGCCATGGCATCGATGACAGGTGGCCAACAGATGGGTTAAGA TGATTGGTGACACTTTTTGTAGAACCCATGTTGTATACAGTTTTCCTTTACTCAGTACAATCTTTTCA') were synthesized by IDT. The sequence encoding a small peptide linker (P-G-A-G-A), inserted between the NRS and the T7 peptide, is highlighted in bold. Gene editing was performed by microinjection of RNA encoding the wild-type Cas9 nuclease (50 ng/µl, TriLink BioTechnologies, #L7606), 25 ng/µl single-guide RNA (sgRNA) targeting exon 4 of *Srsf1*, and 150 ng/µl of both NRS-T7 and WT single-stranded DNA oligonucleotides of a similar length containing 50–100 nucleotides of homology flanking both sides of the sgRNA as repair templates into (C57BL/6 × CBA) F2 zygotes (*Crichton et al., 2017*). The injected zygotes were cultured overnight in KSOM for subsequent transfer to the oviduct of pseudopregnant recipient females (*Joyner, 2000*). From microinjection CRISPR targeting, 57 pups were born, and genomic DNA from ear-clip tissue was subject to PCR screening and subsequent sequence verification of successful mutagenesis events by Sanger sequencing (*Figure 1—figure supplement 1B*). Two heterozygous founder mice were each backcrossed for one generation to C57BL/6 wild-type mice to allow unwanted allelic variants to segregate. Of the resulting 32 pups, 10 animals were confirmed by sequencing to be heterozygous for the correct SRSF1-NRS allele (*Figure 1—figure supplement 1B*, Allele Accession ID: MGI:6452080) and subsequently intercrossed. Genotyping was performed by PCR using the following forward and reverse primers: TTGATGGGCCCAGAAGTCC and ATAGGGCCCTCTAGACAATTTCATCTGTGACAATAGC, respectively.

## Srsf1$^{NRS}$ mice

Two heterozygous $Srsf1^{NRS}$ founder mice were each backcrossed for one generation to C57BL/6 wild-type mice to allow unwanted allelic variants to segregate. Of the resulting 32 pups, 10 animals were confirmed by sequencing to be heterozygous for the correct SRSF1-NRS allele (*Figure 1—figure supplement 1B*, Allele Accession ID: MGI:6452080) and subsequently intercrossed. The phenotypes reported in *Figures 1–3* were present in $Srsf1^{NRS/NRS}$ animals derived from both independent founder mice, confirming their association with homozygosity for $SRSF1^{NRS}$ rather than an off-target or spontaneous mutation. For subsequent studies, $SRSF1^{NRS/+}$ heterozygous animals were crossed onto the ubiquitously expressing ARL13B-Cerulean biosensor $R26Arl13b-Fucci2aR^{Tg/Tg}$ (*Ford et al., 2018*). In this background, $Srsf1^{NRS/NRS}$ mutants were born at the frequencies of 32 $Srsf1^{+/+}$, 75 $Srsf1^{+/NRS}$, 27 $Srsf1^{NRS/NRS}$. Initial genotyping was performed by PCR using the following forward and reverse primers: TTGATGGGCCCAGAAGTCC and ATAGGGCCCTCTAGACAATTTCATCTGTGACAATAGC, respectively. The genotyping of the compound animals was done by TransNetyx.

## Histology

Whole brains from P14 mice were embedded in paraffin wax and 5 µM sections cut using a microtome before oven drying overnight at 60℃. Sections were dewaxed in xylene and rehydrated in decreasing concentrations of ethanol with a final rinse in running water. Sections were stained in hematoxylin for 4 min, rinsed in water, then placed in 1% HCl in 70% EtOH for 5 s before rinsing again in water and treating with lithium carbonate solution for 5 s. Tissue sections were rinsed well in running water for 5 min and stained in eosin for 2 min before rinsing in water and washing three times in 100% EtOH for 1 min each. Sections were cleared in fresh xylene three times for 5 min and mounted using DPX mounting media.

## Heterokaryon assay

This was performed as previously described (*Cazalla et al., 2002*; *Piñol-Roma and Dreyfuss, 1992*), with minor modifications. Donor mouse NSCs were seeded on laminin-coated cover slips, followed by co-incubation with an excess of HeLa cells (recipient) for 3 hr in the presence of 50 µg/ml cycloheximide (Sigma, #C104450), to prevent further protein synthesis in the heterokaryons. The concentration of cycloheximide was then increased to 100 µg/ml, and the cells were incubated for an additional 30 min prior to fusion. Cells were fused with PEG1500 (Roche, #10783641001) in the presence of 100 µg/ml cycloheximide, and the heterokaryons were further incubated for 3 hr in media containing 100 µg/ml. Cells were then fixed and stained with a T7 antibody, and DNA was stained with Hoechst 33258 (Sigma, #63493) to distinguish between mouse and human nuclei.

## Mouse tracheal epithelial cultures

These were established as described previously (*Vladar and Brody, 2013*). $Srsf1^{+/+}$ and $Srsf1^{NRS/NRS}$ animals were always sacrificed as pairs. Tracheas were dissected and dissociated individually. More than five pairs of animals were used to establish tracheal cultures used along this study. Cells were expanded into monolayers of progenitor cells in T25 flasks until confluency and dissociated as described to be seeded onto 6–9 transwells of 6.5 mm each depending on cell density. Under these conditions, our cultures have roughly no ciliary bundles until day 4 and visibly motile cilia until day 7, reaching mature coordinated beating between adjacent bundles by 25–30 days post airlifting. Membranes of transwells were released from inserts and inverted onto a glass-bottom plate for imaging after which cells were harvested for proteomics, immunofluorescence, or electron microscopy.

## Protein extraction, antibodies, and western blotting

Cell pellets were lysed in 50 mM Tris pH 8.0, 150 mM NaCl, 1% NP-40 buffer containing protease inhibitors. Protein samples either from ESCs or liver extracts were separated by SDS-PAGE and electroblotted onto nitrocellulose membranes (Whatman) using iblot System for 6 min (Invitrogen). Nonspecific binding sites were blocked by incubation of the membrane with 5% nonfat milk in PBS containing 0.1% Tween 20 (PBST). Proteins were detected using the following primary antibodies diluted in blocking solution: mouse monoclonal anti-SRSF1 (clone 96; 1:1000; *Hanamura et al., 1998*), mouse monoclonal anti-tubulin (Sigma #T8328), and mouse monoclonal anti-T7 (Novagen, #69522, 1:10,000). Following washing in PBST, blots were incubated with the appropriate secondary

antibodies conjugated to horse-radish peroxidase (Pierce) and detected with Super Signal West Pico detection reagent (Pierce).

## Proteomics

Each transwell (one experimental replicate) of mTECs cultures was lysed in 350 µl of PBS, 2% SDS, and antiproteolytic cocktail, kept at −80°C until all time points were harvested to be run simultaneously. Total mTEC proteomes were derived from two animals/genotype with three experimental replicates per time point (days 4–10, animal pair 1; days 14–18 animal pair two). Testes from P22 or P23 siblings or age matching controls from same breeders were decapsulated and lysed in 500 or 350 µl of PBS, 2% SDS, and anti-proteolytic cocktail, kept at −80°C until four $Srsf1^{+/+}$ and three $Srsf1^{NRS/NRS}$ were collected. In all cases, cell lysates were digested using sequential digestion of LysC (Wako) and trypsin (Pierce) using the FASP protocol (https://pubmed.ncbi.nlm.nih.gov/25063446/). Samples were acidified to 1% TFA final volume and clarified by spinning on a benchtop centrifuge (15k g, 5 min). Sample clean-up to remove salts was performed using C18 stage-tips (*Rappsilber et al., 2003*). Samples were eluted in 25 µl of 80% acetonitrile containing 0.1% TFA and dried using a SpeedVac system at 30°C and resuspended in 0.1% (v/v) TFA such that each sample contained 0.2 µg/ml. All samples were run on an Orbitrap FusionTM Lumos mass spectrometer coupled to an Ultimate 3000, RSL-Nano uHPLC (both Thermo Fisher). 5 µl of the samples were injected onto an Aurora column (25 cm, 75 um ID Ionoptiks, Australia) and heated to 50°C. Peptides were separated by a 150 min gradient from 5–40% acetonitrile in 0.5% acetic acid. Data were acquired as data-dependent acquisition with the following settings: MS resolution 240k, cycle time 1 s, MS/MS HCD ion-trap rapid acquisition, injection time 28 ms. The data were analyzed using the MaxQuant 1.6 software suite (https://www.maxquant.org/) by searching against the murine Uniprot database with the standard settings enabling LFQ determination and matching. The data were further analyzed using the Perseus software suite. LFQ values were normalized, 0-values were imputed using a normal distribution using the standard settings. To generate expression profiles of cilium-related proteins, GO annotations were added and all entries containing 'cilia' were retained. Expression profiles were normalized (using Z-score) and expression profiles were plotted.

## Cilia motility analysis

Cilia motility analysis mTECs were imaged by releasing the membrane from each transwell on which they were grown and inverting it onto a glass-bottom plate. Ciliated bundles were imaged randomly as they appeared on the field of view using a 60× Plan Apochromat VC 1.2 WI DIC N2 lens within a preheated chamber at 37°C. Motile vs. immotile bundles were scored by direct visual inspection of these movies. The number of all counts for each category at each time point was used to calculate the significance of the different proportions between $Srsf1^{NRS/NRS}$ and $Srsf1^{+/+}$ cultures by Fisher's exact test. Other parameters of cilia motility were analyzed in FIJI as indicated in each panel by slowing down animation to manually count beats/s of distinct ciliary bundles or automatically by the custom-written ImageJ plugin 'Cilility_JNH' (available upon request). The underlying analysis method was adapted from *Olstad et al., 2019* and is based on the periodic changes of pixel intensities in the image caused by ciliary beating that are used by the plugin to determine ciliary beat frequency. The intensity time course at each pixel in the image was converted into a frequency spectrum using a Fast Fourier Transformation (code: edu.emory.mathcs.jtransforms.fft by Piotr Wendykier, Emory University). Each spectrum was smoothed with a user-defined averaging sliding window (5 Hz). The position and power of the highest peak in the spectrum within a user-defined range was determined (will be referred to as primary frequency). The user-defined range was defined as 3–62.5 Hz, which represents a quarter of the acquisition frequency (250 Hz). Next, a custom noise-threshold algorithm was applied to separate the image into noise and signal regions: (1) pixels were sorted by the power of the primary frequency; (2) the average and standard deviation (SD) of the 20% of pixel with the lowest power were determined and used to define a power threshold as average + 1.5× SD; and (3) all pixels with a primary frequency power above the threshold were considered as signal pixels, all other pixels were considered as noise pixels. A 'noise' power spectrum was determined as the average + 1.5× SD of all power spectra from pixels belonging to noise regions. The 'noise' power spectrum was subtracted from the power spectrum at all 'signal' pixel positions (power below zero was set to zero). For each cell, the resulting power spectrum from all 'signal' pixels was

averaged in to a 'signal' power spectrum. The frequency of the highest peak in the 'signal' power spectrum within the user-defined range of 3–62.5 Hz determined the ciliary beat frequency. All results were scrutinized by a trained observer.

## Color-coded time projections of swimming mouse sperm

To generate color-coded time projections of time-lapse images of mouse sperm, images were processed as follows in FIJI (Schindelin et al., 2012). They were pre-processed using SpermQ_Preparator (Hansen et al., 2018): (1) pixel intensities were inverted; (2) pixel intensities were rescaled so that intensities in the image covered the whole bit range; (3) the image stack was blurred with a Gaussian blur (sigma = 0.5 px) to reduce noise; (4) a median intensity projection of the image stack was subtracted from all images in the stack to remove the static image background; (5) pixel intensities were again rescaled so that intensities in the image covered the whole bit range; and (6) the background was subtracted using ImageJ's 'Subtract Background' method (radius 5 px). Next, a selected time span (time indicated in the figure legend) was converted into a color-coded time projection using FIJI's function 'Temporal-Color Code' (Schindelin et al., 2012).

## Cell cultures

ESCs were grown on gelatin-coated plates in 2i media: 1:1 neurobasal and Dulbecco's Modified Eagle's Medium (DMEM)/F12, supplemented with $0.5\times$ N2 (Thermo Scientific, #17502048), $0.5\times$ B27 (Thermo Scientific, #17504044), 0.05% BSA (Thermo Scientific, #15260037), 1 mM PD0325901 (Stemcell Technologies, #72182), 3 µM CHIR99021 (Miltenyi Biotec, #130-103-926), 2 mM L-glutamine, 0.15 mM monothioglycerol, 100 U/ml LIF. NSCs were grown on laminin-coated (Sigma Aldrich, #L2020) plates in DMEM/F12 medium supplemented with 2 mM L-glutamine, $0.5\times$ N2, B27, glucose, BSA, HEPES, and 10 ng/ml of both mouse EGF (Peprotech, #315-09) and human FGF-2 (Peprotech, #100-18B). MEFs were cultured in DMEM supplemented with 10% fetal bovine serum (FBS).

## Derivation and maintenance of ESCs

Embryonic stem cells were established from $Srsf1^{NRS/+} \times Srsf1^{NRS/+}$ crosses, following an adapted version of the protocol by Czechanski et al., 2014. E3.5 blastocysts were isolated and plated in 4-well plates pre-seeded with mitomycin-C-treated MEFs. All derivation and downstream propagation of established lines was carried out in 2i media. After the first 48 hr during which the blastocysts were undisturbed, media was changed every other day. Outgrowths were isolated after approximately 1 week and transferred to a fresh feeder-coated plate. Successfully derived cells were slowly weaned from feeders by continual passaging before genotyping to avoid contamination of genomic DNA.

## Neuroectodermal specification and generation of NSCs

ESCs were cultured under feeder-free conditions in 2i medium and were differentiated to NSCs, as previously described with minor modifications (Pollard et al., 2006). Briefly, 1 day prior to induction of differentiation, cells were seeded at high density in 2i medium. The following day cells were detached using Accutase (Stemcell), resuspended in N2B27 media (1:1 neurobasal and DMEM/F12, supplemented with $0.5\times$ N2, $0.5\times$ B27, 0.1 mM 2-mercaptoethanol, 0.2 mM L-glutamine), counted and plated at approximately 10,000 cells per $cm^2$ onto either 15 cm plates or 6-well plates that have been coated with a 0.1% gelatin solution. Culture medium was changed every second day. For derivation of NSCs at day 7 of differentiation, cultures were detached using Accutase, $2–3 \times 10^6$ cells were re-plated into an uncoated T75 flask in NS expansion media, comprising DMEM/F12 medium supplemented with 2 mM L-glutamine, $0.5\times$ N2, B27, glucose, BSA, HEPES, and 10 ng/ml of both mouse EGF and human FGF-2. Within 2–3 days, thousands of cell aggregates formed in suspension culture and were harvested by centrifugation at 700 rpm for 1 min. They were then re-plated onto a laminin coated T75 flask. After few days, cell aggregates attached to the flask and outgrew with NS cell.

## RNA isolation and RT-qPCR

RNA was isolated using TRIzol (Life Technologies, #15596026) or RNAeasy (Qiagen, #74106) following the manufacturer's protocol. RNA was then treated with Turbo Dnase (Ambion, #AM1907) and

transcribed to cDNA using First-Strand Synthesis System from Roche. This was followed by Sybr-Green detection system (Lightcycler 2x SybrGreen Mix, Roche, #04707516001). Testis RNA was transcribed to cDNA using Superscript III and oligoDT. This was followed by RT-qPCR using PrimeTime qPCR probe assays and PrimeTime Gene Expression Master Mix (IDT).

## RNA-seq analysis

RNA was extracted from *Srsf1*[+/+], *Srsf1*[NRS/+], or *Srsf1*[NRS/NRS] MEFs and purified using RNeasy kit from three independent experiments. RNA-seq libraries were generated from Poly(A)[+] mRNA using TrueSeq protocol and sequenced using the Illumina Hi-Seq 4000 machine (WTCRF Edinburgh) to generate 75 bases, paired-end reads. Reads were mapped to the mouse (mm10) genome. Paired reads were pseudoaligned to the GRCm38 (mm10) Ensembl 87 transcriptome using salmon (*Patro et al., 2017*). Splicing changes were inferred from transcript TPMs using SUPPA2 (*Trincado et al., 2018*) from gene definitions in the Ensembl 87. SUPPA2 infers splicing changes (dPSI) from changes in transcript models across the two conditions being compared. These results were filtered on mean transcript expression (TPM > 0.5).

## Cell fractionation and sucrose gradient centrifugation

ESCs, MEFs, or NSCs were treated with 50 µg/ml cycloheximide for 30 min. Cells were subsequently washed twice in ice-cold PBS containing cycloheximide. Cytoplasmic extracts were prepared as previously described (*Sanford et al., 2004*). Testes were homogenized by eight strokes of Dounce homogenizer in the lysis buffer (20 mM Tris-HCl, pH 7.5, 100 mM KCl, 5 mM MgCl$_2$, 1 mM DTT, 0.5% NP-40, 1x protease inhibitor cocktail [EDTA-free], 200 units/ml of RNase inhibitor). Sucrose gradients (10–45%) containing 20 mM Tris, pH 7.5, 5 mM MgCl$_2$, 100 mM KCl were made using the Bio-Comp gradient master. Extracts were loaded onto the gradient and centrifuged for 2.5 hr at 41,000 rpm in Sorvall centrifuge with SW41Ti rotor. Following centrifugation, gradients were fractionated using a BioComp gradient station model 153 (BioComp Instruments, Inc, New Brunswick, Canada) measuring cytosolic RNA at 254 nm. Fractions 1–7 (subpolysomal fraction) and 8–11 (polysomal fractions) were pooled and diluted sucrose concentration adjusted to 20%. The RNA extraction was performed as described above.

## Polysomal shift analysis

Experiments in ESCs, MEFs, and NSCs derived from Srsf1[+/+] and Srsf1[NRS/NRS] mice were performed in three biological replicates. Monosomal, polysomal, and cytoplasmic reads were mapped to the mouse genome sequence (mm10) by STAR software (*Dobin et al., 2013*) (v2.0.7f) with –outFilterScoreMinOverLread 0.3 –out FilterMatchNminOverLread 0.3 settings and Ensembl release M21 transcript annotation. PIs were calculated for each condition for the set of protein coding genes based on a procedure previously described (*Maslon et al., 2014*). The transcript abundance metric, RPKM, was replaced by TPM reads as such normalization allows comparison between samples without biases. The PIs and associated PSRs were calculated for each sample by pooling reads from replicates, as follows.

For each mRNA (*a*), the density of read counts was calculating using the TPM in a given sample (*N*):

$$d(a,N) = \frac{\frac{n(a,N)}{length(a)}}{\sum_{j \in N} \frac{n(j,N)}{length(j)}} 10^6$$

Using these densities, for each transcript and for each of the two samples, polysomal (*poly*) and subpolysomal (*sub*), a polysomal index (*PI*) was defined:

$$PI(a) = \frac{d(a, Npoly)}{d(a, Npoly) + d(a, Nsub)}$$

This index measures the proportion of transcript copies that is present in polysomes.

Then, in order to determine the mRNAs that change polysome occupancy upon NSR, we defined a polysome shift ratio (*PSR*) as the log2 ratio of the PI between the NSR and WT experiments:

$$PSR(a) = \log_2 \frac{PI(a,\ NSR)}{PI(a,\ WT)}$$

The statistical significance of PSRs was assessed by Student's t-test on the variability of PIs of individual replicates. AS events were retrieved using rMATS (v3.2.5) (*Park et al., 2013*). The results were filtered by a minimum coverage of 10 reads per junction and replicate, dPSI > 0.2 and FDR < 0.05. Differential expression analysis was performed with DEseq2 (v1.26.0) (*Love et al., 2014*) and results were filtered by FDR < 0.05. The polysomal-subpolysomal ratio of individual mRNAs in wild-type or SRSF1-NRS expressing testes was quantified by RT-qPCR as described above.

### GO analysis

GO term enrichment analysis was performed using WebGestalt2019 (*Liao et al., 2019*). Genes downregulated at least 0.85-fold (PSR <= -0.23) or upregulated at least 1.15-fold were used as an input. The following parameters were used for the enrichment analysis: minimum number of IDs in the category: 5; and maximum number of IDs in the category: 300.

### Quantification and statistical analysis

All statistical analyses were carried out using GraphPad Prism 8 (version 8.4.1; GraphPad Software, USA) as described in the text. To determine statistical significance, unpaired t-tests were used to compare between two groups, unless otherwise indicated. The mean $\pm$ the standard error of the mean (SEM) is reported in the corresponding figures as indicated. Statistical significance was set at $p<0.05$. Fisher's exact test was used to determine the significance in animal studies and to classify ciliary bundles in motile or immotile during mTEC maturation. All in vitro experiments were repeated in three biological replicates and several litters were used for in vivo studies, as indicated in each section.

## Acknowledgements

We thank Andrew Wood (MRC HGU) for discussion on CRISPR/Cas strategies, Lisa McKie for histology work, and Jimi Wills for mass spectrometry sample preparation and analysis. We are grateful to the MRC Advanced Imaging Resource and CBS animal facility for technical assistance and support throughout the project.

## Additional information

### Funding

| Funder | Grant reference number | Author |
|---|---|---|
| Medical Research Council | MRC Core funding | Pleasantine Mill<br>Ian R Adams<br>Javier F Caceres |
| Deutsche Forschungsgemeinschaft | Germany's Excellence Strategy | Dagmar Wachten |
| Wellcome Trust | Multiuser Equipment 208402/Z/17 | Alex von Kriegsheim |

The funders had no role in study design, data collection and interpretation, or the decision to submit the work for publication.

### Author contributions

Fiona Haward, Magdalena M Maslon, Patricia L Yeyati, Conceptualization, Validation, Investigation, Visualization, Methodology, Writing - original draft, Writing - review and editing; Nicolas Bellora, Data curation, Formal analysis; Jan N Hansen, Alex von Kriegsheim, Validation, Investigation; Stuart Aitken, Data curation, Formal analysis, Validation; Jennifer Lawson, Investigation; Dagmar Wachten, Resources, Investigation; Pleasantine Mill, Ian R Adams, Conceptualization, Supervision, Funding acquisition, Investigation, Writing - original draft, Writing - review and editing; Javier F Caceres,

Conceptualization, Supervision, Funding acquisition, Writing - original draft, Project administration, Writing - review and editing

### Author ORCIDs
Fiona Haward ⓘ https://orcid.org/0000-0001-9048-5600
Magdalena M Maslon ⓘ http://orcid.org/0000-0002-1050-1306
Nicolas Bellora ⓘ http://orcid.org/0000-0001-6637-3465
Jan N Hansen ⓘ https://orcid.org/0000-0002-0489-7535
Dagmar Wachten ⓘ http://orcid.org/0000-0003-4800-6332
Ian R Adams ⓘ https://orcid.org/0000-0001-8838-1271
Javier F Caceres ⓘ https://orcid.org/0000-0001-8025-6169

### Ethics

Animal experimentation: This is stated in the Materials and Methods Section, 'Animal experiments' We followed international, national and institutional guidelines for the care and use of animals. Animal experiments were carried out under UK Home Office Project Licenses PPL 60/4424, PB0DC8431 and P18921CDE in facilities at the University of Edinburgh (PEL 60/2605) and were approved by the University of Edinburgh animal welfare and ethical review body.

### Decision letter and Author response
Decision letter https://doi.org/10.7554/eLife.65104.sa1
Author response https://doi.org/10.7554/eLife.65104.sa2

# Additional files

### Supplementary files
• Transparent reporting form

### Data availability

Total RNA-seq data related to splicing analysis have been deposited in GEO under accession code GSE157269G. Polysomal, monosomal and cytoplasmic RNA-sequencing data have been deposited in GEO under accession code GSE161828. The mass spectrometry proteomics data is presented as LFQ values in the form of Excel tables in Figure 4-source data 1 and in Figure 4-figure supplement 1-source data 1. The complete raw datasets can be downloaded from ProteomeXchange Consortium via the PRIDE partner repository with the dataset identifier PXD019859. Cilia motility data analyzed in Fiji using the custom-written ImageJ plugin "Cilility_JNH" was deposited in zenodo (https://doi.org/10.5281/zenodo.5138072).

The following datasets were generated:

| Author(s) | Year | Dataset title | Dataset URL | Database and Identifier |
|---|---|---|---|---|
| Haward F, Maslon MM, Yeyati PL, Bellora N, Aitken S, von Kriegsheim A, Mill P, Adams IR, Caceres JF | 2021 | Translational activity of the splicing factor SRSF1 is required for development and cilia function | https://www.ebi.ac.uk/pride/archive/projects/PXD019859 | PRIDE, PXD019859 |
| Haward F, Maslon MM, Yeyati PL, Bellora N, Aitken S, von Kriegsheim A, Mill P, Adams IR, Caceres JF | 2021 | Translational activity of the splicing factor SRSF1 is required for development and cilia function | https://www.ncbi.nlm.nih.gov/geo/query/acc.cgi?acc=GSE157269 | NCBI Gene Expression Omnibus, GSE157269 |
| Haward F, Maslon MM, Yeyati PL, Bellora N, Aitken S, von Kriegsheim A, | 2021 | Identification of polysomal-association changes upon ablation of cytoplasmic SRSF1 | https://www.ncbi.nlm.nih.gov/geo/query/acc.cgi?acc=GSE161828 | NCBI Gene Expression Omnibus, GSE161828 |

| | | | | |
|---|---|---|---|---|
| Mill P, Adams IR, Caceres JF | | | | |
| Hansen JN | 2021 | hansenjn/Cilility_JNH _release of Version v0.2.2 (v0.2.2) | https://doi.org/10.5281/zenodo.5138072 | Zenodo, 10.5281/zenodo.5138072 |

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
