## [Decision Letter]

**Acceptance summary:**

This study presents the unexpected finding that a well known and broadly expressed splicing regulatory protein SRSF1 plays a specialized role in the assembly of motile cilia. Using an interesting application of mouse genetics, the authors show that motile cilia are disrupted by the selective loss of cytoplasmic SRSF1, and further that a set ciliary protein mRNAs require SRSF1 for proper translation. These findings connect ciliary biology and RNA biology through a novel mechanism of posttranscriptional regulation.

**Decision letter after peer review:**

Thank you for submitting your article "Nucleo-cytoplasmic shuttling of splicing factor SRSF1 is required for development and cilia function" for consideration by *eLife*. Your article has been reviewed by 3 peer reviewers, and the evaluation has been overseen by Douglas Black as Reviewing Editor and James Manley as the Senior Editor. The following individuals involved in review of your submission have agreed to reveal their identity: Michaela Müller-McNicoll (Reviewer #1); Maxence V Nachury (Reviewer #2); Florian Heyd (Reviewer #3).

The reviewers have discussed the reviews with one another and the Reviewing Editor has drafted this decision to help you prepare a revised submission.

Summary:

This study from Haward and colleagues examines the cytoplasmic function of the SR protein SRSF1. SRSF1 is well studied for its role in exon recognition during pre-mRNA splicing. The protein is also known to accompany some mature mRNAs to the cytoplasm where it can affect translation and other processes. The authors create mice carrying a carrying a chimeric mutant SRSF1 allele containing the nuclear retention sequence from the non-shuttling SR protein SRSF2. They show in cultured cells that this modified SRSF1 remains nuclear and no longer shuttles to the cytoplasm. Mice homozygous for this SRSF1 mutation do not exhibit the early embryonic lethal phenotype seen with a SRSF1 null allele, but instead are born in Mendelian ratios albeit with significant postnatal growth defects. These include defects such as hydrocephalus and loss of sperm motility that are associated with disruption of ciliary function. EM analyses of mouse tracheal epithelial cells (mTEC) revealed abnormal microtubule organization within the ciliary axonemes. These results indicate that the SRSF1 mutation is causing misfunction of motile cilia. RNAseq analyses of MEFs and NSCs identified minimal changes in splicing as a result of the mutation. Moderate changes in cytoplasmic levels of mRNAs were observed, interpreted as limited defects in mRNA export. More striking changes were observed in mRNA polysome indices indicating changes in the translation of particular mRNAs. Down-regulated proteins were enriched for genes in the motile cilium and other functional categories. Finally, the authors examined ciliary function in mutant and wildtype mTEC cells. By a variety of measurements, they show that the SRSF1 mutation leads to delayed assembly of cilia, changes in ciliary beat frequency and amplitude, and decreased levels of ciliary proteins.

The reviewers all considered these findings to be of significant interest, both regarding the function of cytoplasmic SRSF1 and the implication of posttranscriptional regulation in ciliary gene expression. The mouse model and the quality of the data are also strengths. However, several of the conclusions are not sufficiently supported by the presented data to constitute a complete study. Additional analyses are needed to solidify both the findings regarding translational control and the link to defects in motile cilia.

Essential Revisions:

1. A major disconnect is that the defect is seen in multi-ciliated cells, but the RNA and protein analyses are done on ESC, MEFs, and NSC. Additional RNA and protein analyses are needed in mTECs to conclude that the proposed mechanism of translational inhibition is occurring in the relevant cell types.

2. Another substantive concern is whether the developmental and cellular defects seen in the SRSF1NRS/NRS mice are largely due to changes in translation. Figure 4 S1C shows that the export of GAPDH is decreased 2-fold in ESC. More concerning is the *increase* in nuclear export of DNAAF3 and DNAAF1 in the mutant NSC. It is not clear how this analysis was done. Are they simply measuring cytoplasmic read counts or measuring cytoplasmic to nuclear ratios? It would be helpful to present in the main body of the paper a comparison of the RNA export ratio and the polysome shift ratio for the known motile cilia proteins. Without additional data and better presentation, one is unable to conclude that the observed defects are directly caused by decreases in the translation of mRNAs involved in motile cilia biogenesis.

3. SRSF1 expresses isoforms with three different C-termini. The inserted NRS of the mutant allele will not be present in two of these isoforms. These isoforms are likely differentially expressed in different tissues. It thus needs to be shown that the chimeric protein is expressed, and which isoforms are prevalent, in the tissues relevant to this study, e.g. brain, sperm or tracheal epithelium cultures.

4. There are several alternative explanations of the data that need at least to be discussed if not ruled out experimentally. SRSF1 is thought to bind to multiple mRNAs in the nucleus that it accompanies to the cytoplasm. By forcing the SRSF1 to remain nuclear, has the inserted SRSF2 NRS created an aberrant neomorphic phenotype by keeping certain RNAs and perhaps other associated shuttling factors nuclear? Alternatively, SRSF2 shuttling has been shown to depend on the differentiation state (Botti et al., 2017). The fact that the mice survive early development and only show strong phenotypes after birth allows the interpretation that the SRSF2 NRS actually shuttles during early development, but no longer in terminally differentiated cells. Similarly, is it known whether cells may also differ in wildtype SRSF1 shuttling? Is it possible that other shuttling SR proteins compensate for the lack of SRSF1 in mRNA export, but they fail to compensate its role in translation? These alternative models also bear some discussion.

5. The authors should comment on overlap between the identified translational targets and the available iCLIP data on SRSF1 binding sites. Are some of the translational targets seen to be directly bound by SRSF1? Do the SRSF1 binding sites correlating with translational control show common locations in UTRs or exons? Similarly, what is the overlap between the proteomic analysis in Figure 5 and the polysome profiling in Figure 4?

6. Many of the experiments and methods are not sufficiently explained. How exactly is the polysome index calculated? What exactly is the "cilia cell cycle biosensor" mouse strain and what exactly is it used to measure? What protocol was used to generate NSCs? Many smaller changes to the figures and legends are suggested below.

---

## [Author Response]

Essential Revisions:1. A major disconnect is that the defect is seen in multi-ciliated cells, but the RNA and protein analyses are done on ESC, MEFs, and NSC. Additional RNA and protein analyses are needed in mTECs to conclude that the proposed mechanism of translational inhibition is occurring in the relevant cell types.

The reviewers were right in raising this point. The reason that we carried out the RNA and protein analyses on ESC, MEFs, and NSC, was to be able to compare undifferentiated vs differentiated cells. Use of these cellular systems also allowed us to obtain sufficient material to carry out, splicing, mRNA export and polysomal shift analysis. We obtained important conclusions showing that the absence of SRSF1 from the cytoplasm in these cell lines correlates with significant changes in mRNA translation in NSCs, and to a lesser extent in MEFs and ESCs. Interestingly, we observed that a nuclear restricted SRSF1 does not compromise pre-mRNA splicing and is associated with relatively few changes in mRNA export. Overall, these studies highlight that cytoplasmic activities of SRSF1 are mainly related to mRNA translation with perhaps some minor contribution of mRNA export. More importantly, these data also suggested that SRSF1-dependent translation becomes more prominent in differentiating cell types, consistent with a role of SRSF1 as a translational enhancer during specific cellular demands, as opposed to being a constitutive component of the translational machinery.

The sizeable amount of input needed for these translational studies are currently technically not possible from our mTEC cultures. However, we have instead followed the suggestion of the reviewers and addressed the role of SRSF1 in mRNA translation during motile ciliogenesis, using testes as a source of material. The use of testes is relevant due the shared molecular components responsible for cilia/flagellar movement and to the overlapping motility phenotypes observed between sperm flagella and ciliated tracheal cells in Srsf1^NRS/NRS^ mice (Figure 3C-E). These experiments have been challenging in COVID times where our mouse colonies were severely reduced as a precaution for lockdown interruptions. For these experiments, the required generation of multiple litters to obtain sufficient male Srsf1^NRS /NRS^ mice (1:4 * 1:2 = 1/8 pups) to obtain sufficient material to carry out a consistent analysis.

In addition, we present total proteomic analysis of mouse testes during the first wave of spermatogenesis at the onset of assembly of the motile flagellar machinery, post-natal days 22-23 (New panel H in Figure 4 and panels E and F in Figure 4—figure supplement 1). This revealed a decreased abundance of ciliary motor-related proteins in Srsf1^NRS/NRS^ testes that largely overlapped with those downregulated in Srsf1^NRS/NRS^ tracheal cultures. But more importantly data in New Figure 6, clearly demonstrates that the reduced abundance of ciliary proteins observed in the proteomic analysis is due to lower translation of these ciliary mRNAs, as shown by their decreased association with polysomes observed in polysomal shift analysis of mouse testes.

We believe that this is a very important result that conclusively shows that the absence of cytoplasmic SRSF1 results in defects of mRNA translation, linked to transcripts related to motile ciliary function in the relevant tissue.

Altogether, our original data in NSCs, combined with these new experiments in mouse testes and the spectrum of cilia-related phenotypes strongly suggest a prominent role for the cytoplasmic activities of SRSF1 in the biology of motile cilia. Whereas our data points out to a role of SRSF1-mediated translation, we cannot rule out other cytoplasmic effects that may contribute to the phenotypes observed.

2. Another substantive concern is whether the developmental and cellular defects seen in the SRSF1NRS/NRS mice are largely due to changes in translation. Figure 4 S1C shows that the export of GAPDH is decreased 2-fold in ESC. More concerning is the increase in nuclear export of DNAAF3 and DNAAF1 in the mutant NSC. It is not clear how this analysis was done. Are they simply measuring cytoplasmic read counts or measuring cytoplasmic to nuclear ratios? It would be helpful to present in the main body of the paper a comparison of the RNA export ratio and the polysome shift ratio for the known motile cilia proteins. Without additional data and better presentation, one is unable to conclude that the observed defects are directly caused by decreases in the translation of mRNAs involved in motile cilia biogenesis.

Whereas we cannot conclusively rule out SRSF1 cytoplasmic functions other than mRNA translation having a role in the phenotypes observed, we observed a very large effect of the SRSF1-NRS allele in mRNA translation in NSCs and in testes (Figure 5 and Figure 6, respectively).

To clarify the data presented in Figure 5—figure supplement 1 (previously Figure 4), the export analysis was done by measuring cytoplasmic to nuclear ratios for selected mRNAs. We do not believe that observing changes in the export of some mRNAs is concerning. Indeed, the observed modest increase in nuclear export of Dnaaf3 and Dnaaf1 might reflect some feedback mechanism due to decreased translation of these two mRNAs in SRSF1-NRS-expressing NSCs (PSR of -0.51 and -1.35, respectively). Further, although we did not measure mRNA export in testes, should export of Dnaaf3 and Dnaaf1 be also increased in this tissue, it would reinforce the impact of SRSF1 on their translation as the respective protein levels as well as their polysome association are significantly lower in Srsf1^NRS/NRS^ animals than those of sibling controls (Figures 5 and 6). We cannot rule out that other cytoplasmic functions are also affected, and we are showing the examples of mRNAs whose export is affected. We show that whilst export of only a small group of RNAs is affected, several hundred mRNAs are affected at the level of translation. In addition, it needs to be stressed out that the polysomal shift values are normalized to the levels of available mRNAs in the cytoplasm and are therefore not compounded by mRNA export but are instead an absolute measure of mRNA translation.

3. SRSF1 expresses isoforms with three different C-termini. The inserted NRS of the mutant allele will not be present in two of these isoforms. These isoforms are likely differentially expressed in different tissues. It thus needs to be shown that the chimeric protein is expressed, and which isoforms are prevalent, in the tissues relevant to this study, e.g. brain, sperm or tracheal epithelium cultures.

The reviewers raised the issue of alternative isoforms for SRSF1. Indeed, the Krainer lab thoroughly analyzed the expression and regulation of these isoforms (Sun et al., (2010) NSMB | PMID: 20139984). In this study the authors characterized six alternatively spliced SRSF1 mRNA isoforms and showed that the major isoform encodes full-length protein, whereas the others are either retained in the nucleus or degraded by nonsense-mediated mRNA decay. Quoting from their paper “isoforms III–VI undergo excision of one or two introns in their 3′ UTRs, resulting in PTCs that should trigger NMD”

We introduced the NRS in the major predominant form of SRSF1, so we do not think that the lack of NRS in the putative minor SRSF1 isoforms will in any way affect our conclusions. This would be a concern, if no molecular or mouse phenotypes were observed; one could reason that the lowly expressed isoforms could still shuttle and compensate. This does not seem to be the case. Furthermore, WB analysis using an anti-SRSF1 antibody reveals a single band, corresponding to the tagged-SRSF1-NRS protein, as well as the endogenous SRSF1 protein in NSCs, trachea and testes (modified Figure 1D), as well as in mousederived ESCs, liver, brain and testes (new Figure 1—figure supplement 2).

4. There are several alternative explanations of the data that need at least to be discussed if not ruled out experimentally. SRSF1 is thought to bind to multiple mRNAs in the nucleus that it accompanies to the cytoplasm. By forcing the SRSF1 to remain nuclear, has the inserted SRSF2 NRS created an aberrant neomorphic phenotype by keeping certain RNAs and perhaps other associated shuttling factors nuclear?

Our data shows that there are no major changes in mRNA export in the mutant mice (Figure 5—figure supplement 1, panels B and C), yet we have seen extensive changes in mRNA translation (Figure 5B). So, although we cannot rule out this hypothesis, but we think that it is extremely unlikely.

Furthermore, all papers to date examining the role of SR proteins as export adaptors have relied on knockouts or knock-downs of individual SR proteins. Under those circumstances depletion of an individual SR protein will abolish or decrease its interactions with the export adaptor NXF1 in the nucleus, consequently abrogating mRNA export. In our current study, we do not alter the total levels of SRSF1, rather we abrogate its transit and localization in the cytoplasm (SRSF1-NRS). Under these conditions, we do not expect the interaction of SRSF1 with NXF1 in the nucleus to be affected.

While the reviewer/s suggestion that ‘the inserted SRSF2 NRS may keep certain RNAs and associated shuttling factors nuclear’ is a possibility, there is no evidence in the literature to indicate that this is the case. Indeed, experimental data from the Fu lab contradicts this hypothesis, since they postulate that there is a release of non-shuttling SR proteins from the exported mRNA and exchange with shuttling SR proteins (Lin et al., (2005) Mol Cell PMID: 16285923). Quoting from this paper:

“To understand whether shuttling is obligatory for individual shuttling SR proteins to function in vivo and why nonshuttling SR proteins do not interfere with nuclear export of postsplicing mRNPs, we analyzed mRNP maturation in MEFs complemented by various mutant SR proteins. We observed that shuttling and nonshuttling SR proteins are segregated during mRNP maturation in an orderly fashion”

“While shuttling SR proteins remain associated with mRNPs to facilitate mRNA export, nonshuttling SR proteins are sorted away from postsplicing mRNPs, perhaps during an early cotranscriptional splicing step”

Based on this, we would support the idea that SRSF1-NRS is released from target mRNAs in the nucleus and does not interfere with export of target mRNAs. Indeed, this sorting process is influenced by SR protein phosphorylation and in that study the Fu laboratory used the same SRSF1-NRS construct upon which this mouse was engineered (provided by our laboratory) and showed that SRSF1-NRS remains associated with a nuclear matrix-attached insoluble fraction and remained hyperphosphorylated. We would expect the endogenous protein to behave similarly.

Alternatively, SRSF2 shuttling has been shown to depend on the differentiation state (Botti et al., 2017). The fact that the mice survive early development and only show strong phenotypes after birth allows the interpretation that the SRSF2 NRS actually shuttles during early development, but no longer in terminally differentiated cells.

We are fully aware of this work, which focuses on differential SRSF2 shuttling in pluripotency, which is an extremely transient window of pre-implantation development. This is cited in the Discussion (page 17, lines 388 and 406).

Similarly, is it known whether cells may also differ in wildtype SRSF1 shuttling? Is it possible that other shuttling SR proteins compensate for the lack of SRSF1 in mRNA export, but they fail to compensate its role in translation? These alternative models also bear some discussion.

The only difference in SRSF1 shuttling has been provided in the Botti et al. paper, related to the state of cell differentiation (please see above). Again, we could not rule that other shuttling SR proteins compensate for the lack of SRSF1 in mRNA export and/or translation. However, our data showing global reduction of mRNA translation in NSCs and of candidate cilia mRNAs in testes are entirely consistent with a predominant role for SRSF1 in the cytoplasm in mRNA translation, which does not seem to be compensated by the cytoplasmic presence of other shuttling SR proteins. Furthermore, the role of SRSF1 in mRNA export is via binding of nuclear adaptors, which occurs in the nucleus and is most likely not affected in the SRSF1-NRS mice. This is entirely different from previous studies that either knocked-out or knocked-down expression of individual SR proteins. This will affect the nuclear phase of mRNA export. In our experimental design, we do not alter the nuclear distribution of SRSF1, we simply prevent its export to the cytoplasm. Please see discussion above related to the paper by the Fu laboratory demonstrating an exchange of non-shuttling for shuttling proteins in mRNPs.

5. The authors should comment on overlap between the identified translational targets and the available iCLIP data on SRSF1 binding sites. Are some of the translational targets seen to be directly bound by SRSF1? Do the SRSF1 binding sites correlating with translational control show common locations in UTRs or exons?

This is an excellent suggestion but unfortunately CLIP datasets are mostly available for human cells. There are not SRSF1 CLIP datasets available for mESC or mouse NSCs. We used CLIP data from MEFs; we found that all SRSF1 translational targets contained at least one CLIP-tag. However, we did not see an enrichment for CLIP tags for mRNA affected vs not changing. In order to performed comprehensive analysis of SRSF1 binding sites, CLIP experiments in the relevant cell lines are required. We believe that this is beyond the scope of this manuscript.

Similarly, what is the overlap between the proteomic analysis in Figure 5 and the polysome profiling in Figure 4?

We now have added both proteomic analysis and polysome profiling in a same tissue, mouse testes (see New Figure 4H, Figure 4—figure supplement 1, panels E and F, and Figure 6).

6. Many of the experiments and methods are not sufficiently explained. How exactly is the polysome index calculated?

We are now providing more details on how polysome index was calculated (see Polysomal shift analysis in the Methods section).

What exactly is the "cilia cell cycle biosensor" mouse strain and what exactly is it used to measure?

The cilia cell cycle biosensor is our reporter animal that allows the simultaneous characterization of cell cycle progression with cilia dynamics and morphometrics (Ford et al., 2018). For this study, we have used this reporter only for its ciliary membrane associated protein ARL13B fused to a fluorescent tag to allow for the direct visualisation of live cilia from all explanted tissues. Crossing Srsf1-NRS alleles onto this background expedited the studies, as it immediately revealed that Srsf1^NRS/NRS^ animals presented grossly normal numbers of multiciliated cells in the brain and choroid plexus. We had grand designs to use this Srsf1; Fucci2-Arl13b animals to do in-depth cell cycle studies given roles for SRSF1 in translational control of mitotic spindle and cell cycle but these studies are ongoing and outside the scope of this study.

What protocol was used to generate NSCs? Many smaller changes to the figures and legends are suggested below.

As for the protocol to generate NSCs, we believe that this is described in great detail in the Methods section and reflects well how NSCs were generated. We have added a reference to clarify this further (Pollard et al., 2006 PMID: 17141035).